
# Entanglement spectrum and symmetries in non-Hermitian fermionic non-interacting models

Loïc Herviou[1*], Nicolas Regnault[2, 3], Jens H. Bardarson[1]

**1** Department of Physics, KTH Royal Institute of Technology, Stockholm, 106 91 Sweden
**2** Joseph Henry Laboratories and Department of Physics,
Princeton University, Princeton, New Jersey 08544, USA
**3** Laboratoire de Physique de l'École normale supérieure, ENS, Université PSL, CNRS,
Sorbonne Université, Université Paris-Diderot, Sorbonne Paris Cité, Paris, France.

⋆ herviou@kth.se

## Abstract

We study the properties of the entanglement spectrum in gapped non-interacting non-Hermitian systems, and its relation to the topological properties of the system Hamiltonian. Two different families of entanglement Hamiltonians can be defined in non-Hermitian systems, depending on whether we consider only right (or equivalently only left) eigenstates or a combination of both left and right eigenstates. We show that their entanglement spectra can still be computed efficiently, as in the Hermitian limit. We discuss how symmetries of the Hamiltonian map into symmetries of the entanglement spectrum depending on the choice of the many-body state. Through several examples in one and two dimensions, we show that the biorthogonal entanglement Hamiltonian directly inherits the topological properties of the Hamiltonian for line gapped phases, with characteristic singular and energy zero modes. The right (left) density matrix carries distinct information on the topological properties of the many-body right (left) eigenstates themselves. In purely point gapped phases, when the energy bands are not separable, the relation between the entanglement Hamiltonian and the system Hamiltonian breaks down.

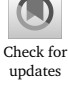

# 1 Introduction

Topology has become one of the main aspects of condensed matter physics over the last few decades [1–6]. The classification of topological phases led to numerous advances in the understanding of electronic condensed matter and to a plethora of new resilient phenomena [7–12]. One of the core principles of topology in condensed matter physics is the bulk-boundary correspondence [6, 13, 14]: topological properties in the bulk of the system lead to the appearance of particular edge states at its boundaries. As these states originate from bulk properties, they are resilient to local perturbations that do not change the topological classification of the system—for instance by breaking the relevant symmetries. This bulk-boundary correspondence also affects the entanglement properties of the different eigenstates, and in particular the ground state, of the Hamiltonian.

Entanglement has proved to be an efficient probe of many-body physics. Entanglement entropy scaling laws are for example able to discriminate between different universality classes of gapless phases, in particular in one dimension [15–18], but also can include terms that have a topological origin and characterize the fundamental topological excitations of the system [19, 20]. Of relevance to this work is the notion of the entanglement Hamiltonian—the logarithm of the reduced density matrix of a subpart of the total system—and its eigenspectrum, the entanglement spectrum [21–24]. Due to the bulk-boundary correspondence, if the selected subsystem does not break any symmetry, the entanglement Hamiltonian in a topological system has similar properties and edge states as the original Hamiltonian with open

boundary conditions, even when starting from a periodic system [21, 25–27]. As such, it has been a remarkably useful tool to characterize topological systems.

Non-Hermitian Hamiltonians are an extension of standard quantum mechanics that describe dissipative systems in a minimalistic fashion. Instead of considering density matrix evolutions such as Lindblad's equations, dissipation is represented as non-Hermitian terms that either give a finite life-time or amplify the different eigenstates of the Hamiltonian [28]. Numerous experiments have been realized, showcasing the many differences between these systems and their Hermitian counterparts [29–38]. Similarly, the extension of the topological concepts developed for Hermitian quantum mechanics to these new systems has been a fruitful field of research [39]. Symmetry-based applications have been proposed [40–43], but several notions are still actively discussed—the bulk-boundary correspondence being one [39,44–55]. Indeed, the phase diagram of the same model can vary significantly depending on the choice of boundary conditions (open or periodic), a phenomenom dubbed the non-Hermitian skin effect. The correspondence can actually be redefined in two different ways: One can redefine an effective Brillouin zone for the periodic Hamiltonian where the momentum can take complex values [48,56]; the topological invariants computed on this new Brillouin zone are then in agreement with the phase diagram of the open system. Conversely, the correspondence can be based on the singular value decomposition (SVD) of the Hamiltonian instead of the eigenvalue decomposition [40,42,43,57]. The SVD-based phase diagrams of the open and periodic systems coincide, and topological phases are characterized by the presence of edge-localized singular zero modes.

In this article, we study the entanglement spectrum in non-Hermitian systems and its relation to the topology of the original Hamiltonian, as a first step towards a better understanding of non-Hermitian topology in many-body physics. After a quick reminder of the properties of the density matrix and the entanglement Hamiltonian in Hermitian systems, we propose two complementary definitions of the density matrix, depending on whether we want to focus on the biorthogonal interpretation of non-Hermitian quantum mechanics [58], or if we are more interested in the structure of the right or left eigenstates of the Hamiltonian. We also show that Wick's theorem and Peschel's formula [59] are still valid in non-Hermitian systems which allows us to efficiently compute the entanglement spectrum of free fermionic theories. We then discuss the different symmetries that can protect the topology of non-Hermitian Hamiltonians, and how they translate into symmetries of the reduced density matrix and the entanglement Hamiltonian depending on the choice of many-body state. In particular, for right density matrices, symmetries of the Hermitian entanglement Hamiltonian might differ from the symmetries of the non-Hermitian system Hamiltonian, leading to a different topological classification of the former. After briefly introducing the non-Hermitian Su-Schrieffer-Heger (SSH) model [48, 60–65], we use it to exemplify how and when the entanglement spectrum inherits topological properties from the original Hamiltonian. We find that when bands can be separated, the biorthogonal entanglement Hamiltonian perfectly reproduces the physics of the corresponding periodic system Hamiltonian, with the presence of singular and energy edge modes accurately predicted by the bulk topological invariants. The right entanglement Hamiltonian describes the topology of the right eigenstates themselves, and its classification differs from the system Hamiltonian due to the emergence of different symmetries. Finally, we verify that our results are also valid on a variety of two-dimensional models.

## 2 Density matrices and entanglement spectrum in non-Hermitian systems

In this Section, we discuss the possible definitions of a density matrix in a non-Hermitian setting. Let us introduce the following notation: We denote by $\mathcal{H}$ the many-body Hamiltonian and assume it can be diagonalized, i.e, it has only $1 \times 1$ Jordan blocks,

$$\mathcal{H} = \sum_n \mathcal{E}_n \left| \psi_n^R \right\rangle \left\langle \psi_n^L \right|, \text{ with } \left\langle \psi_n^L \middle| \psi_m^R \right\rangle = \delta_{m,n}. \tag{1}$$

$\left| \psi_n^R \right\rangle$ ($\left\langle \psi_n^L \right|$) are the right (left) eigenvectors of the many-body Hamiltonian. Any many-body state $\left| \phi^R \right\rangle$ for such system can be decomposed into the eigenstates $\left| \psi_n^R \right\rangle$, i.e., $\left| \phi^R \right\rangle = \sum_n \phi_n \left| \psi_n^R \right\rangle$. We define the corresponding left vector $\left| \phi^L \right\rangle \propto \sum_n \phi_n \left| \psi_n^L \right\rangle$. For convenience, in the rest of this paper, we always take the following normalization convention:

$$\left\| \left| \phi^R \right\rangle \right\|^2 = 1 \text{ and } \left\langle \phi^L \middle| \phi^R \right\rangle = 1. \tag{2}$$

In this paper, we focus on non-interacting fermionic models such that

$$\mathcal{H} = \vec{c}^\dagger H \vec{c}. \tag{3}$$

$\vec{c}^\dagger = (c_1^\dagger, ..., c_N^\dagger)$ is a vector of $N$ fermionic creation operators satisfying the usual anticommutation algebra

$$\{c_i^\dagger, c_j\} = \delta_{i,j}, \ \{c_i, c_j\} = 0. \tag{4}$$

$H$ is the single particle Hamiltonian that can be diagonalized as

$$H = \sum_n E_n \left| R_n \right\rangle \left\langle L_n \right|, \tag{5}$$

with $\left\langle L_n \middle| R_m \right\rangle = \delta_{m,n}$ and $\left\langle R_n \middle| R_n \right\rangle = 1$.

We define $d_{n,R}^\dagger$ ( $d_{n,L}^\dagger$ ) as the creation operator related to the one-body eigenstate $\left| R_n \right\rangle$ ($\left| L_n \right\rangle$):

$$d_{n,R}^\dagger = \sum_j \left\langle j | R_n \right\rangle c_j^\dagger. \tag{6}$$

They satisfy the modified fermionic anticommutation rule:

$$\{d_{m,R}^\dagger, d_{n,L}\} = \delta_{m,n}, \ \{d_{m,R}^\dagger, d_{n,R}^\dagger\} = \{d_{m,L}, d_{n,L}\} = 0. \tag{7}$$

The other anticommutators do not have a simple expression.

### 2.1 Density matrices

In Hermitian systems, the density matrix describing a system is the positive-definite Hermitian operator $\rho$ that verifies that the expectation value of any observable $\mathcal{O}$ is given by

$$\langle \mathcal{O} \rangle = \text{Tr}(\rho \mathcal{O}), \tag{8}$$

where $\langle \ . \ \rangle$ is the expectation value. If the system is in a pure state $\left| \phi \right\rangle$, the density matrix $\rho$ is simply the projector $\left| \phi \right\rangle \left\langle \phi \right|$, while a thermal state is given by $\rho = Z^{-1} \exp(-\beta \mathcal{H})$, with $Z = \text{Tr}[\exp(-\beta \mathcal{H})]$. The time evolution of $\rho$ is given by the Heisenberg equation (we set $\hbar = 1$)

$$i \frac{d\rho}{dt} = [\mathcal{H}, \rho]. \tag{9}$$

The reduced density matrix $\rho_{\mathcal{A}}$ characterizing the state of a subsystem $\mathcal{A}$ can be obtained from $\rho$ by taking the partial trace over all degrees of freedom not in $\mathcal{A}$:

$$\rho_{\mathcal{A}} = \mathrm{Tr}_{\overline{\mathcal{A}}}\, \rho. \tag{10}$$

In non-Hermitian systems, the difference between left- and right- eigenstates leads to different possible definitions of the density matrix. This definition choice depends on which properties we want to preserve or emphasize, even for a pure state. We focus in this paper on static properties, but we will mention some of the dynamical properties.

Following the biorthogonal interpretation of non-Hermitian quantum mechanics [58], observables are computed using both the left- and right- states of a system:

$$\langle \mathcal{O} \rangle_{RL} = \left\langle \phi^L \middle| \mathcal{O} \middle| \phi^R \right\rangle. \tag{11}$$

This naturally leads to the biorthogonal density matrix

$$\rho^{RL} = \left| \phi^R \right\rangle \left\langle \phi^L \right|. \tag{12}$$

The reduced density matrices can be obtained from Eq. (10), and the Heisenberg equation is left unchanged. The trace of $\rho^{RL}$ is conserved during time evolution. On the other hand, $\rho^{RL}$ is neither Hermitian nor positive-definite.

If we consider instead a more conventional approach where non-Hermitian systems are effective models for dissipative dynamics without quantum jumps [66–70], the average values of observables are given by

$$\langle \mathcal{O} \rangle_R = \left\langle \phi^R \middle| \mathcal{O} \middle| \phi^R \right\rangle. \tag{13}$$

The natural density matrix is therefore the right density matrix

$$\rho^R = \frac{\left| \phi^R \right\rangle \left\langle \phi^R \right|}{\mathrm{Tr}\, \left| \phi^R \right\rangle \left\langle \phi^R \right|}. \tag{14}$$

By convention, we take $\left| \phi^R \right\rangle$ to be of norm 1 such that $\mathrm{Tr}\, \left| \phi^R \right\rangle \left\langle \phi^R \right| = 1$. Equation (10) is still valid, and $\rho^R$ and all associated reduced density matrices are Hermitian positive-definite operators. $\rho^R$ then satisfies the equation [71]

$$i\frac{d\rho^R}{dt} = H\rho^R - \rho^R H^\dagger - \rho^R \mathrm{Tr}\left(H\rho^R - \rho^R H^\dagger\right). \tag{15}$$

Enforcing the constraint $\mathrm{Tr}\, \rho^R = 1$ leads to non-linearity in the time evolution of $\rho^R$. If $\left| \phi^R \right\rangle$ is a right eigenstate, then $\rho^R$ is constant. We denote by $\rho^L$ the equivalent density matrix replacing right by left vectors.

## 2.2 Entanglement spectrum

The entanglement Hamiltonian $\mathcal{H}_E$ of a subsystem $\mathcal{A}$ is given by

$$\rho_{\mathcal{A}} = \exp(-\mathcal{H}_E). \tag{16}$$

The entanglement spectrum of $\rho$ is the spectrum of $\mathcal{H}_E$. When the total system is in a pure state and we use $\rho^R$ as the density matrix, the entanglement spectrum of $\rho_{\mathcal{A}}^R$ is directly related to the Schmidt decomposition of $\left| \phi^R \right\rangle$. Indeed, the Schmidt decomposition writes as:

$$\left| \phi^R \right\rangle = \sum_n \lambda_n \left| \phi^R_{n,\mathcal{A}} \right\rangle \otimes \left| \phi^R_{n,\overline{\mathcal{A}}} \right\rangle, \tag{17}$$

where $\lambda_n > 0$ and $\{\left|\phi_{n,\mathcal{A}}^R\right\rangle\}$ ($\{\left|\phi_{n,\overline{\mathcal{A}}}^R\right\rangle\}$) is a set of orthonormal vectors of $\mathcal{A}$ ($\overline{\mathcal{A}}$) satisfying

$$\left\langle \phi_{m,\mathcal{A}}^R | \phi_{n,\mathcal{A}}^R \right\rangle = \left\langle \phi_{m,\overline{\mathcal{A}}}^R | \phi_{n,\overline{\mathcal{A}}}^R \right\rangle = \delta_{m,n}. \tag{18}$$

Due to the orthogonality conditions,

$$\rho_{\mathcal{A}} = \mathrm{Tr}_{\overline{\mathcal{A}}} \left|\phi^R\right\rangle\left\langle\phi^R\right| = \sum_n \lambda_n^2 \left|\phi_{n,\mathcal{A}}^R\right\rangle\left\langle\phi_{n,\mathcal{A}}^R\right|, \tag{19}$$

and consequently, the eigenvalues $\Xi_n$ of $\mathcal{H}_E$ are nothing but $-2\log\lambda_n$. For the biorthogonal density matrix $\rho^{RL}$, there is no simple relation between the Schmidt decomposition of the eigenvectors and the eigenvalues of the entanglement Hamiltonian.

If $\mathcal{H}_E = \vec{c}^{\dagger} H_E \vec{c} + z\mathrm{Id}$, $z \in \mathbb{C}$, the reduced density matrix is a generalized fermionic Gaussian state [72] ($z$ is a irrelevant normalization factor that will not be discussed in the following). The eigenvalues $\xi_n$ of $H_E$ form the single particle entanglement spectrum, and its eigenvectors the entanglement modes. In the rest of the paper, as we only discuss such Gaussian states, we refer to $\xi_n$ and $H_E$ as the (single-particle) entanglement spectrum and Hamiltonian.

## 3 Entanglement spectrum of Gaussian states and Wick's theorem

In Ref. [59], Peschel derived a technique to efficiently compute the entanglement spectrum of eigenstates of quadratic Hermitian Hamiltonian (Slater determinants) or of Gaussian density matrices. It can be summarized as follows: any correlation function for such states can, according to Wick's theorem, be obtained from a combination of two-fermion correlation functions. Moreover, computing the correlation functions restricted to any subsystem $\mathcal{A}$ only requires two-fermion correlators restricted to that subsystem. Let $C$ be the two-site correlation matrix defined by $C_{i,j} = \left\langle c_j^{\dagger} c_i \right\rangle$ in such a state, and $C_{\mathcal{A}}$, the restriction of $C$ to the subsystem $\mathcal{A}$. $C_{\mathcal{A}}$ can be diagonalized into

$$C_{\mathcal{A}} = \sum_{n=1}^{N_{\mathcal{A}}} s_n \left|R_n^{\mathcal{A}}\right\rangle\left\langle R_n^{\mathcal{A}}\right| \text{ with } 0 \le s_n \le 1. \tag{20}$$

$N_{\mathcal{A}}$ is the number of fermionic modes in $\mathcal{A}$. The Gaussian state defined through Eq. (16) with the (single-particle) entanglement Hamiltonian $H_E = \sum_n \xi_n \left|R_n^{\mathcal{A}}\right\rangle\left\langle R_n^{\mathcal{A}}\right|$ with $\xi_n = \ln(s_n^{-1} - 1)$ gives the same correlation matrix $C_{\mathcal{A}}$. Note that if $s_n = 0$ or $1$, $\xi_n$ is formally $-\infty$ or $+\infty$. In practice, this limiting case does not occur as long as $\mathcal{A}$ is not the entire system, though the smallest and largest values of $s_n$ get exponentially close to the extrema with increasing system size. Since the Gaussian state also satisfies Wick's theorem, all fermionic correlators have the same expectation value whether using $\rho_{\mathcal{A}}$ or the above Gaussian state. Therefore, necessarily,

$$\rho_{\mathcal{A}} = \exp(-\vec{c}^{\dagger} H_E \vec{c}), \tag{21}$$

and the entanglement spectrum can be directly obtained from the eigenvalues of the reduced correlation matrix, which can be computed polynomially in system size.

To apply a similar trick to non-Hermitian systems, we need first to verify that Wick's theorem applies to both formulation of density matrices in Eqs. (12) and (14), as well as to non-Hermitian Gaussian states. Secondly, we should verify that fermionic Gaussian states generate all possible non-Hermitian correlation matrices.

We start with the biorthogonal density matrix $\rho^{RL}$ and Wick's theorem. We consider eigenstates of the Hamiltonian that can be written as $|\phi_R\rangle = \prod_n d_R^{\dagger s_n}|0\rangle$, with $s_n = 0$ or $1$. The

corresponding left-eigenstate is $|\phi_L\rangle = \prod_n d_L^{\dagger s_n} |0\rangle$. In the biorthogonal case, straightforward algebra mapping $c^\dagger$ to $d_R^\dagger$ and $c$ to $d_L$ leads to

$$C^{RL} = \sum_n s_n |R_n\rangle \langle L_n|, \text{ where } C_{i,j}^{RL} = \text{Tr}(c_j^\dagger c_i \rho^{RL}), \tag{22}$$

which has eigenvalues 0 or 1, i.e., the occupation numbers are the eigenvalues of $C^{RL}$. $|R_n\rangle$ (resp. $\langle L_n|$) are the right (resp. left) eigenstates of the single-particle Hamiltonian $H$. This mapping also offers a proof of Wick's theorem: once expressed in the correct left and right basis, the correlators of the non-Hermitian system behave exactly as if the system was Hermitian. Similarly, non-Hermitian Gaussian states of the form $\rho = e^{-\vec{c}^\dagger H_E \vec{c}}$ also verify Wick's theorem; if $H_E$ is diagonalizable, this follows trivially from the Hermitian case. By continuity of the matrix exponentiation and the trace, it is also true for non-diagonalizable $H_E$.

Now we need to prove that all non-Hermitian correlation matrices also admit a Gaussian antecedent. In App. A, we exhibit the antecedent of any correlation matrix that forms a single Jordan block of arbitrary size. The generalization to arbitrary correlation matrix is straightforward. Similarly to the Hermitian case, eigenvalues 0 or 1 of the correlation matrix correspond to divergent energies for the Gaussian states. If the correlation matrix is diagonalizable, the corresponding entanglement Hamiltonian is also diagonalizable, and its eigenmodes are the eigenvectors of the correlation matrix. If the correlation matrix is not diagonalizable, the entanglement Hamiltonian $H_E$ is also not diagonalizable. It follows naturally from App. A that the correlation matrix and the entanglement Hamiltonian have the same Jordan block structure (same number of Jordan blocks of the same size). Matching Jordan blocks in the correlation matrix and the entanglement Hamiltonian act on the same eigenspace. The canonical basis of this eigenspace that leads to the Jordan form will generically be different in the two matrices.

When considering the right density matrix $\rho^R$, it is convenient to work in an orthonormalized basis of the occupied states. Let $(i_1, ... i_m)$ be the indices of the occupied modes, with $m$ the number of occupied states. Further let $\mathcal{Q} = (|Q_1\rangle, ..., |Q_m\rangle)$ be an orthonormal basis of $\text{Span}(|R_{i_1}\rangle, ..., |R_{i_m}\rangle)$ and

$$q_j^\dagger = \sum_j \langle j|Q_j\rangle c_j^\dagger, \tag{23}$$

such that

$$|\phi^R\rangle = \prod_{j=1}^m q_j^\dagger |0\rangle. \tag{24}$$

We can complete $\mathcal{Q}$ into an orthonormal basis of the single particle space. $|\phi^R\rangle$ is then the ground state of the Hermitian Hamiltonian $\mathcal{H}' = \sum_{j=m+1}^N q_j^\dagger q_j - \sum_{j=1}^m q_j^\dagger q_j$. From this follows that $\rho^R$ verifies Wick's theorem and that its reduced density matrices are Hermitian Gaussian states. Finally, the correlation matrix can be efficiently obtained from the eigenvalue decomposition of $H$. Let $P_m = \sum_{n=1}^m |R_{i_n}\rangle \langle n|$ be the $N \times m$ matrix of occupied states, with $|n\rangle$ an orthonormal basis of $\mathbb{C}^m$. The matrix $Q = \sum_{n=1}^m |Q_{i_n}\rangle \langle n|$ is obtained from the $QR$ decomposition of $P$ and $C = QQ^\dagger$.

Both definitions of the density matrices lead to Gaussian reduced density matrices. We can efficiently compute the two-site correlation matrix from the diagonalization of the single-site

Hamiltonian, and thus the entanglement spectrum.

$$\xi_n = \log\left[s_{\mathcal{A},n}^{-1} - 1\right],\tag{25}$$

where $s_{\mathcal{A},n}$ is an eigenvalue of the correlation matrix $C_{\mathcal{A}}$ restricted to the subsystem $\mathcal{A}$ we consider. Note that the formula (25) is the same as in the Hermitian case. Since the entanglement Hamiltonian might have complex eigenvalues, the entanglement spectrum is only defined modulo $2i\pi$. We will choose the phases such that the symmetries of the correlation matrix are respected. If $C_{\mathcal{A}}$ is diagonalizable, the left and right entanglement modes are its left and right eigenvectors.

## 4 Symmetries and entanglement Hamiltonian

Symmetries play a fundamental role in the behavior of the entanglement spectrum in Hermitian systems [25, 73]. A natural prescription to study topological effects on the entanglement spectrum for symmetry-protected topological phases is to select a (ground) state that does not break any of the protecting symmetries. The correlation matrix, and by extension all reduced density matrices, will have the same symmetries, and the entanglement Hamiltonian can potentially be in the same topological phase as the initial one. In this section we demonstrate that this prescription is still natural in the non-Hermitian case. More precisely, we discuss the effects of symmetries on the correlation matrix and reduced density matrices, in relation with the band structure of the eigenvalues. Indeed, two types of gaps can be defined in non-Hermitian systems [43], as illustrated in Fig. 1. The system is said to be point gapped if it possesses no eigenvalues in the neighborhood of a single point of the complex energy plane, usually $E = 0$, as depicted in Fig. 1(a). In sharp contrast with the (anti-)Hermitian case, bands need not be separable. Conversely, the system is said to be line gapped if there exists a one-dimensional manifold in the complex energy plane with no eigenvalues in its neighborhood, separating the energies into two sets or bands, as shown in Fig. 1(b-c). Due to symmetry, this manifold is generally either the real or the imaginary axis. An Hamiltonian then admits a real line gap if the real part of its eigenvalues is gapped in the Hermitian meaning of the word. Depending on the type of gap, Hamiltonians will have different topological classification and the obtained correlation matrices will have different symmetries.

### 4.1 Conserved quantities

Let $\mathcal{O}$ be an operator that commutes with $\mathcal{H}$. Then $\mathcal{O}$ and $\mathcal{H}$ preserve each other's left and right eigenspaces, and eigenspaces of $\mathcal{H}$ can be labeled by the eigenvalues $o$ of $\mathcal{O}$. Let $\mathcal{A}$ be a part of the system such that $\mathcal{O} = \mathcal{O}_{\mathcal{A}} + \mathcal{O}_{\overline{\mathcal{A}}}$, with $\mathcal{O}_{\mathcal{A}}$ ($\mathcal{O}_{\overline{\mathcal{A}}}$) acting only on $\mathcal{A}$ (the rest of the system). Using Schmidt decomposition, one can write any eigenstate of $\mathcal{H}$ as

$$\left|\psi^{R/L}, o\right\rangle = \sum_{o_{\mathcal{A}}+o_{\overline{\mathcal{A}}}=o} \sum_n \lambda^n_{o_{\mathcal{A}},o_{\overline{\mathcal{A}}}} \left|\psi^{R/L}_{\mathcal{A},n}, o_{\mathcal{A}}\right\rangle \otimes \left|\psi^{R/L}_{\overline{\mathcal{A}},n}, o_{\overline{\mathcal{A}}}\right\rangle.\tag{26}$$

As $\left\langle\psi^L_{\overline{\mathcal{A}},m}, o_{\overline{\mathcal{A}}}\middle|\psi^R_{\overline{\mathcal{A}},n}, o'_{\overline{\mathcal{A}}}\right\rangle = \delta_{o_{\overline{\mathcal{A}}},o'_{\overline{\mathcal{A}}}}\delta_{m,n}$, the biorthogonal reduced density matrix is:

$$\rho^{RL} = \sum_{o_{\mathcal{A}}} \sum_n \left(\sum_{o_{\overline{\mathcal{A}}}} |\lambda^n_{o_{\mathcal{A}},o_{\overline{\mathcal{A}}}}|^2\right) \left|\psi^R_{\mathcal{A},n}, o_{\mathcal{A}}\right\rangle\left\langle\psi^L_{\mathcal{A},n}, o_{\mathcal{A}}\right|.\tag{27}$$

The reduced density matrix therefore commutes with $\mathcal{O}_{\mathcal{A}}$, whose eigenvalues are still good quantum numbers.

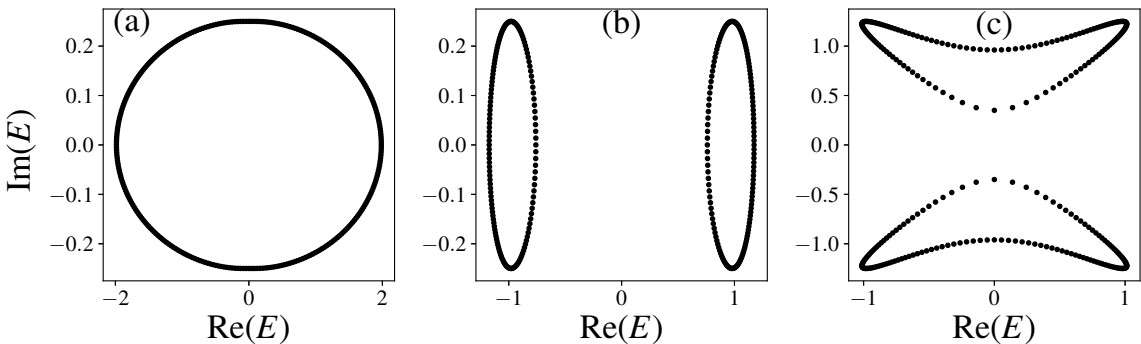

Figure 1: Three different types of band structure for non-Hermitian two-level systems. In (a) the band structure is non-separable while still having a point gap. It is a fundamentally non-Hermitian structure, which admits both purely imaginary and real energies. Only (b) and (c) have well-defined line gap. This line gap can be the imaginary (b) or the real (c) axis, which relates naturally to a Hermitian (b) or an anti-Hermitian (c) limit. Purely imaginary (real) energies are then forbidden.

Now we turn to the right density matrix. Schmidt decomposition applied to each $(o_{\mathcal{A}}, o_{\overline{\mathcal{A}}})$ sector ensures that $\left\langle \psi^R_{\mathcal{A},m}, o_{\overline{\mathcal{A}}} \middle| \psi^R_{\mathcal{A},n}, o_{\overline{\mathcal{A}}} \right\rangle = \delta_{m,n}$. If the eigenspaces of $\mathcal{O}$ are orthogonal (for example if $\mathcal{O}$ is a normal operator, i.e., $\mathcal{O}^\dagger \mathcal{O} = \mathcal{O}\mathcal{O}^\dagger$ or Hermitian), then $\left\langle \psi^R_{\mathcal{A},m}, o_{\overline{\mathcal{A}}} \middle| \psi^R_{\mathcal{A},n}, o'_{\overline{\mathcal{A}}} \right\rangle$ is zero if $o_{\overline{\mathcal{A}}} \neq o'_{\overline{\mathcal{A}}}$. The reduced density matrix is then given by

$$\rho^R = \sum_{o_{\mathcal{A}}} \sum_n \left( \sum_{o_{\overline{\mathcal{A}}}} |\lambda^n_{o_{\mathcal{A}}, o_{\overline{\mathcal{A}}}}|^2 \right) \left| \psi^R_{\mathcal{A},n}, o_{\mathcal{A}} \right\rangle \left\langle \psi^R_{\mathcal{A},n}, o_{\mathcal{A}} \right|. \tag{28}$$

It also commutes with $\mathcal{O}_{\mathcal{A}}$ and the symmetry is preserved. On the other hand, if $\mathcal{O}$ is not normal, then its eigenspaces are no longer orthogonal and $\mathcal{O}_{\mathcal{A}}$ a priori does not commute with the right reduced density matrix. The $\mathcal{O}$ symmetry is then broken in the entanglement Hamiltonian.

## 4.2   $\mathbb{Z}_2$ (anti-)unitaries symmetries for biorthogonal density matrices $\rho^{RL}$

We now focus on the $\mathbb{Z}_2$ unitary and anti-unitary symmetries used in topological classification of Hermitian and non-Hermitian Hamiltonian. Four types of symmetries have been proposed to classify non-Hermitian Hamiltonians through the Bernard-LeClair symmetry classes [74–77]:

$$Ch : H = -u_c H u_c^\dagger, \text{ with } u_c u_c^\dagger = I,\ u_c^2 = I,$$

$$T_{\varepsilon_t} : H = \varepsilon_t u_t H^* u_t^\dagger, \text{ with } u_t u_t^\dagger = I,\ u_t u_t^* = \eta_t I,$$

$$P_{\varepsilon_p} : H = \varepsilon_p u_p H^T u_p^\dagger, \text{ with } u_p u_p^\dagger = I,\ u_p u_p^* = \eta_p I,$$

$$PH_{\varepsilon_{ph}} : H = \varepsilon_{ph} u_{ph} H^\dagger u_{ph}^\dagger, \text{ with } u_{ph} u_{ph}^\dagger = I,\ u_{ph}^2 = I,$$

where the $\varepsilon$'s and $\eta$'s can take values $\pm 1$. $Ch$ is a chiral symmetry, $T$ and $P$ are two flavors of particle-hole ($\varepsilon = -1$) or time-reversal ($\varepsilon = 1$) symmetries and $PH$ is pseudo-hermiticity. All unitary transformations ($u_c$, $u_t$, $u_p$ and $u_{ph}$) are required to be compatible with the subsystem $\mathcal{A}$: If the correlation matrix $C$ verifies some symmetry relations, there exists reduced unitaries defined on $\mathcal{A}$ such that $C_{\mathcal{A}}$ also satisfies the same relation.

Table 1: The symmetry conditions for both the Hamiltonian and the two-site correlation matrix. The first column is the symmetry verified by the Hamiltonian. The second column marks how energies appear in pairs (e.g., $(E_n, -E_n)$ means that energies appear in pairs of opposite signs). The third column is the symmetry transformation obeyed by the correlation matrix, while the fourth summarizes the corresponding conditions on the occupancy numbers of the many-body state. The table can be interpreted in two ways. Starting from a symmetric Hamiltonian, the fourth column indicates the constraints on the occupancy numbers of the many-body state such that the entanglement Hamiltonian also admits the same symmetries. Conversely, starting from a Gaussian state with a symmetric $H_E$, the third column indicates the symmetries verified by the correlation matrix.

| $H$ symm | $E_n$ | $C$ | $s_n$ |
|---|---|---|---|
| $Ch$ | $(E_n, -E_n)$ | $u_c C u_c^\dagger + C = I$ | $s_n + s_{-n} = 1$ |
| $T_+$ | $(E_n, E_n^*)$ | $u_t C^* u_t^\dagger = C$ | $s_n = s_{n^*}^*$ |
| $T_-$ | $(E_n, -E_n^*)$ | $u_t C^* u_t^\dagger + C = I$ | $s_n + s_{-n^*}^* = 1$ |
| $P_+$ | None | $u_p C^T u_p^\dagger = C$ | None |
| $P_-$ | $(E_n, -E_n)$ | $u_p C^T u_p^\dagger + C = I$ | $s_n + s_{-n} = 1$ |
| $PH_+$ | $(E_n, E_n^*)$ | $u_{ph} C^\dagger u_{ph}^\dagger = C$ | $s_n = s_{n^*}^*$ |
| $PH_-$ | $(E_n, -E_n^*)$ | $u_{ph} C^\dagger u_{ph}^\dagger + C = I$ | $s_n + s_{-n^*}^* = 1$ |

For simplicity, we now assume that $H$ has no degenerate eigenvalues. We use the shorthand notations $|R_{n^*}\rangle$ for the eigenvector associated to $E_n^*$ and $|R_{-n}\rangle$ to $-E_n$, and similarly for all related quantities. $\left|R_n^*\right\rangle$ is the complex conjugate of $|R_n\rangle$.

Depending on the state we consider, a symmetry in the Hamiltonian can translate into two different symmetries on the correlation matrix, and therefore on the entanglement Hamiltonian. Here we discuss explicitly the case of the pseudo-Hermitian $PH_-$ symmetry, the other cases following straightforwardly.

The symmetry on the Hamiltonian translates into

$$u_{ph} |R_n\rangle = e^{i\alpha_n} \mathcal{N}_n |L_{-n^*}\rangle, \tag{29}$$

$$u_{ph} |L_n\rangle = e^{i\alpha_n} \mathcal{N}_n^{-1} |R_{-n^*}\rangle, \tag{30}$$

with eigenvalues coming in pairs $(E_n, -E_n^*)$. For simplicity, we skip for now the case of purely imaginary energies. $e^{i\alpha_n}$ is a complex phase and $\mathcal{N}_n$ is the normalization constant $|| |L_n\rangle ||^{-1}$. Following Eq. (22), we obtain

$$u_{ph} C^\dagger u_{ph}^\dagger = \sum_n s_n^* |R_{-n^*}\rangle \langle L_{-n^*}| . \tag{31}$$

If $s_n^* + s_{-n^*} = 1$, we obtain

$$u_{ph} C^\dagger u_{ph}^\dagger + C = 1. \tag{32}$$

This relation can be satisfied by simply occupying the states with negative (or positive) real part of the energy in the many-body state we consider. Such a choice coincides with the conventional choice of the ground state for Hermitian systems with particle-hole symmetry at half-filling, and is a consistent choice if the Hamiltonian admits a real line gap as in Fig. 1(b). Correspondingly, if an entanglement Hamiltonian verifies the $PH_-$ symmetry, it will satisfy

Eq. (32). Conversely, up to the $2i\pi$ degrees of freedom in the definition of entanglement energy, assuming there are no degeneracies, if the correlation matrix verifies Eq. (32), the entanglement Hamiltonian is necessary $PH_-$ symmetric. Another interesting relation emerges if we take $s_n^* = s_{-n^*}$. In a Hermitian system, such a condition makes very little physical sense: it attributes the same occupancy to states with opposite energies. In the non-Hermitian case, it cannot be rejected a priori. If the spectrum has an imaginary line gap, such as shown in Fig. 1(c), selecting the band with either positive or negative imaginary part results in such a relation. In other words, it corresponds to the natural occupation of the anti-Hermitian limit of the Hamiltonian. The correlation matrix then satisfies

$$u_{ph} C^\dagger u_{ph}^\dagger = C, \tag{33}$$

which is the $PH_+$ symmetry. Similarly, the corresponding entanglement Hamiltonian will have the same $PH_+$ symmetry, with eigenvalues coming in pairs $(\xi_n, \xi_n^*)$.

Finally, let us discuss the case of purely real or imaginary eigenmodes. If the Hamiltonian $H$ admits some purely imaginary eigenvalues, then $u_{ph}$ maps the right eigenvectors to the corresponding left eigenvectors if there are no degeneracies. Then, Eq. (32) cannot be satisfied by any of the eigenstates of $\mathcal{H}$ as it requires $s_n + s_{-n^*}^* = 1$. The $PH_-$ symmetry is spontaneously broken. On the other hand, such a mode is still compatible with the emergent $PH_+$ symmetry. If the Hamiltonian now has purely real eigenvalues, then the relation $s_n^* = s_{-n^*}$ requires to attribute the same occupancy to states with opposite energies, which is generally unphysical when studying half-filling properties. When the Hamiltonian has both purely real and imaginary eigenenergies, for example for the non-separable bands shown in Fig. 1a, then there is no natural choice of many-body state that leads to a surviving symmetry in the entanglement Hamiltonian. Note that in finite systems, picking adequate boundary conditions and system sizes can prevent the symmetry breaking, as we will exemplify in Secs. 6.1.1 and 7.2.

Such a change of the symmetry representation occurs for most of previously considered symmetries. In Table 1, we summarize the required conditions on the many-body state occupancies in order to have the exact same symmetry in the system Hamiltonian and the entanglement Hamiltonian. These conditions are generically compatible with (and natural in) the Hermitian limit. In each case, the corresponding entanglement Hamiltonian will have the same symmetry as the Hamiltonian if $C$ and $s_n$ satisfy the indicated relation, and therefore the energy pair constraint is also valid for the entanglement Hamiltonian. In Table 2, we summarize the required conditions to have the previously described change in the symmetry representation. With the exceptions of the $Ch$ and $P_-$ symmetries, these conditions would be natural in the anti-Hermitian limit of the Hamiltonian. The choice of the more physically relevant many-body state depends on the band structure of the original Hamiltonian.

## 4.3 $\mathbb{Z}_2$ (anti-)unitaries symmetries for right density matrices $\rho^R$

We now turn to the right density matrices and investigate how symmetries of the system Hamiltonian can map to the entanglement Hamiltonian. Some non-Hermitian symmetries relate left and right eigenvectors of the Hamiltonian, while only the latter are involved in the computation of the density matrix and the associated correlation matrix. Additionally, the right eigenvectors do not form an orthogonal basis, which also affect some symmetry relations. Let us consider here the example of group BDI$^\dagger$ [43] (group 14 in Ref. [42]), characterized by the presence of the symmetries $P_+$, $T_-$ and $PH_-$. In itself, this group is topologically trivial in dimension 1. The symmetries enforce the following relations on the eigenvectors of the

Table 2: Conditions to obtain an alternate symmetry representation in the entanglement Hamiltonian of the different symmetries of the system Hamiltonian. The original symmetry of the Hamiltonian (first column), under the suitable choice of many-body state (second column) leads to different symmetry properties for the correlation matrix (third column), which means that $H_E$ will have a different symmetry (last column). Two special cases emerge. The chiral symmetry leads to the appearance of a new conserved quantity, corresponding to the chiral operator $u_c$. The $P_+$ symmetry does not appear in this table. Under the assumption that there are no degeneracies in $H$, the entanglement Hamiltonian is also always $P_+$ symmetric. It is interesting to note that the $P_-$ symmetry then leads to a doubly degenerate entanglement Hamiltonian. Except from $Ch$ and $P_-$, these symmetry conditions are natural in the anti-Hermitian limit.

| $H$ sym. | $s_n$ | $C$ | $H_E$ sym. |
|---|---|---|---|
| $Ch$ | $s_n = s_{-n}$ | $[u_c, C] = 0$ | |
| $T_+$ | $s_n + s_{n*}^* = 1$ | $u_t C^* u_t^\dagger + C = I$ | $T_-$ |
| $T_-$ | $s_n = s_{-n*}^*$ | $u_t C^* u_t^\dagger = C$ | $T_+$ |
| $P_-$ | $s_n = s_{-n}$ | $u_p C^T u_p^\dagger = C$ | $P_+$ |
| $PH_+$ | $s_n + s_{n*}^* = 1$ | $u_{ph} C^\dagger u_{ph}^\dagger + C = I$ | $PH_-$ |
| $PH_-$ | $s_n = s_{-n*}^*$ | $u_{ph} C^\dagger u_{ph}^\dagger = C$ | $PH_+$ |

Hamiltonian (assuming no energy degeneracies):

$$
\begin{aligned}
P_+: \quad |R_n\rangle &= \mathcal{N}_n e^{i\alpha_n^p} u_p |L_n^*\rangle, \\
|L_n\rangle &= \mathcal{N}_n^{-1} e^{i\alpha_n^p} u_p |R_n^*\rangle, \\
T_-: \quad |R_n\rangle &= e^{i\alpha_n^t} u_t |R_{-n*}^*\rangle, \\
|L_n\rangle &= e^{i\alpha_n^t} u_t |L_{-n*}^*\rangle, \\
PH_-: \quad |R_n\rangle &= \mathcal{N}_{-n*} e^{i\alpha_n^{ph}} u_{ph} |L_{-n*}\rangle, \\
|L_n\rangle &= \mathcal{N}_{-n*}^{-1} e^{i\alpha_n^{ph}} u_{ph} |R_{-n*}\rangle,
\end{aligned}
$$

with $\mathcal{N}_n$ the normalization factor $|||L_n\rangle||^{-1}$ and the $e^{i\alpha}$'s are complex phases. Let us start with $PH_-$ and consider a state where all modes with negative real part of the energy are occupied. We assume that there are no purely imaginary modes. As eigenvalues come in pairs $(E_n, -E_n^*)$, the system is at half-filling and the corresponding biorthogonal density matrix verifies all three symmetries. Let $Q = \{|Q_n\rangle\}_n$ be the Schmidt orthonormalization of the family of occupied modes introduced in Section 3. By construction $Q$ spans half the single-particle Hilbert space. The set $u_{ph}Q$ is orthogonal to $Q$ as $\langle R_m|L_n\rangle = \delta_{m,n}$ (using $u_{ph}^2 = I$) and is also an orthonormal family as $u_{ph}$ is unitary. It is therefore the orthogonal complement of $Q$ such that $(Q, u_{ph}Q)$ forms a complete basis of the single-particle Hilbert space. The right correlation matrix associated to this eigenstate is

$$
C^R = \sum_n |Q_n\rangle \langle Q_n|, \tag{34}
$$

and consequently $C^R$ is chiral symmetric:

$$
C^R + u_{ph} C^R u_{ph}^\dagger = \sum_n |Q_n\rangle \langle Q_n| + \sum_n u_{ph} |Q_n\rangle \langle Q_n| u_{ph}^\dagger = I. \tag{35}
$$

Table 3: Summary of how the different non-Hermitian symmetries of the Hamiltonian can induce the standard Atland-Zirnbauer [78] symmetries on the right entanglement Hamiltonian. The first column lists the non-Hermitian symmetry, the second column indicates the induced Hermitian symmetry, and in the last column, we present the required conditions on the many-body state (expressed in the occupancy of the different eigenmodes of the initial Hamiltonian). It is interesting to note that the pseudo Hermitian symmetry $PH_-$ (resp. $PH_+$) requires that the spectrum has no purely imaginary (resp. real) eigenvalues in the absence of spectrum degeneracies.

| $nH$ sym. | $H$ sym | Condition on occupancies |
|-----------|---------|--------------------------|
| $P_-$ | PHS | $s_n + s_{-n} = 1$ |
| $T_+$ | TRS | $s_n = s_{n^*}$ |
| $T_-$ | TRS | $s_n = s_{-n^*}$ |
| $PH_-$ | Chiral | $s_n + s_{-n^*} = 1$ |
| $PH_+$ | Chiral | $s_n + s_{n^*} = 1$ |

On the other hand, let us consider the effect of $P_+$ on the same state. $u_p Q$ is also an orthonormal family, but it is a priori neither orthogonal to $Q$ nor generated by it, and we obtain no special relation on the density matrix. In this state, the $P_+$ (and therefore also the $T_-$ symmetry) is broken as it actually maps the right density matrix to the left. If there are no additional symmetries, the right-density matrix then falls into the Hermitian $AI$ symmetry class, which is topologically non-trivial in one dimension.

As we have seen, only considering either the right or left density matrices might lead to radically different symmetry properties of the entanglement Hamiltonian, and thus reveal different properties of the system Hamiltonian. In the presence of $PH_-$, the natural choice of many-body eigenstate can lead to the emergence of a chiral symmetry in the right-density matrix, even though it is not present in the original Hamiltonian. The additional chiral symmetry may lead to topological signatures and features in the entanglement hamiltonian and consequently in left and right eigenstates of the original Hamiltonian even though the Hamiltonian is in principle trivial.

This result is similar but not equivalent to the line-gap classification obtained in Ref. [43]. In particular, while the $T_-$ and $P_+$ symmetries are relevant to the line gap classification, they only map the right density matrix $\rho^R$ to the left density matrix $\rho^L$, which does not put any strong constraints on $\rho^R$ itself, and therefore does not constrain its topological properties. For example, in the case of $T_+$, $T_-$ and $Ch$ symmetry (group $AI + S_+$), the line gap classification predicts a $\mathbb{Z}$ topological invariant while the right density-matrix is only $T_+$ symmetric and therefore topologically trivial according the standard Hermitian classification. In Table 3, we summarize how the different non-Hermitian symmetries can transform into a symmetry in the right entanglement Hamiltonian, and the conditions on the many-body states in order for such a symmetry to exist.

This potential discrepancy between the topological properties of the entanglement Hamiltonian and of the system's Hamiltonian is in particular relevant when studying dissipative trajectories with post-selection [66–70]. The post-selection allows us to simplify the Lindblad evolution into a purely non-Hermitian Hamiltonian problems, and the density matrix of the system is exactly the right density matrix that we consider. While the topological properties of the Hamiltonian still matter as far as the existence of zero-modes are concerned [70], the existence of topologically stable observables will be governed by the properties of the right eigenvectors only.

# 5 The non-Hermitian SSH chain

The non-Hermitian Su-Schrieffer-Heeger [48, 60–65] (SSH) model is an extension of the celebrated SSH model with additional non-Hermitian terms. Its Hamiltonian reads

$$\mathcal{H} = -(t_1 + \gamma) \sum_j c_{j,B}^\dagger c_{j,A} - (t_1 - \gamma) \sum_j c_{j,A}^\dagger c_{j,B}$$
$$- t_2 \sum_j \left( c_{j+1,A}^\dagger c_{j,B} + c_{j,B}^\dagger c_{j+1,A} \right) + i\mu \sum_j (n_{j,A} - n_{j,B}) . \quad (36)$$

$t_1$ ($t_2$) is an intra- (inter-) unit-cell coupling, $\gamma$ is a non-reciprocal contribution to the hopping, and $\mu$ encodes alternating losses and gains. $j$ denotes the unit-cell while $A/B$ is the sublattice index. We consider a system of $L$ unit cells. In the following, we denote with $\sigma^\alpha$ with $\alpha = x, y, z$ the Pauli operators acting on the sublattice degrees of freedom. In the rest of the paper, we assume for simplicity $t_1, t_2, \mu, \gamma \geq 0$ and fix our energy scale to $t_2 = 1$.

The non-Hermitian SSH model possesses topological and trivial phases that are directly connected to the corresponding phases in the Hermitian SSH model. More saliently, it hosts a topological phase specific to non-Hermitian models. When $\gamma \neq 0$, it exhibits the so-called non-Hermitian skin-effect [39, 44–52], i.e., a break-down of the conventional bulk-boundary correspondence of topological systems. The eigenvalues and eigenvectors of the system with open boundary conditions (OBC) strongly differ from the ones of the system with periodic boundary conditions (PBC). Consequently, the conventionnal phase diagram—where a phase transition is characterized by the closing of the gap in the energy spectrum—depends on the choice of boundary conditions. With OBC, eigenstates tend to localize towards one of the boundary of the system. On the other hand, the singular value phase diagram — where a phase transition is based on the closing of the gap in the singular value decomposition of the single-particle Hamiltonian $H$ — does respect the bulk-boundary correspondence. We summarize here the phase diagram and the main properties of the model.

The PBC phase diagram can be easily computed and is shown in Fig. 2. In the chiral limit $\mu = 0$ [48, 64, 65], the Hamiltonian is time-reversal $T_+$ symmetric with $u_t = \text{Id}$, particle-hole $T_-$ symmetric with $u_t = \sigma^z$ and chiral $Ch$ symmetric. It falls in the non-Hermitian AI+$S_+$ [43] class (group 36 in Ref. [42]), with two $\mathbb{Z}$ topological invariants. Several formulations have been proposed for these invariants [40, 43, 49, 57, 64, 79, 80]. In this paper we use

$$\nu_+ = \frac{i}{2\pi} \int\limits_{\text{BZ}} \text{Tr}(Q_k^\dagger \partial_k Q_k), \quad (37)$$

$$\nu_- = \frac{i}{2\pi} \int\limits_{\text{BZ}} \text{Tr}(\sigma^z Q_k^\dagger \partial_k Q_k), \quad (38)$$

where $BZ$ is the Brillouin zone and $Q_k$ is the singular-flattened Hamiltonian [57] at momentum $k$. Namely, if the singular value decomposition of the Bloch Hamiltonian $H_k$ associated to the single-particle counterpart of $\mathcal{H}$ in Eq. (36) is $H_k = U_k \Lambda_k V_k^\dagger$, with $\Lambda_k$ a positive diagonal matrix and $U_k$ and $V_k$ two unitary matrices, then

$$Q_k = U_k V_k^\dagger. \quad (39)$$

The phase "H-Topo" (resp. "nH-Topo") has non trivial winding number and is characterized by two (resp. a single) zero singular values when the system is open . "H-Topo" is adiabatically connected to the Hermitian topological phase, while "nH-Topo" is purely non-Hermitian, with (point-)gapped energy bands that are nonetheless non-separable. "H-Triv" is connected to the

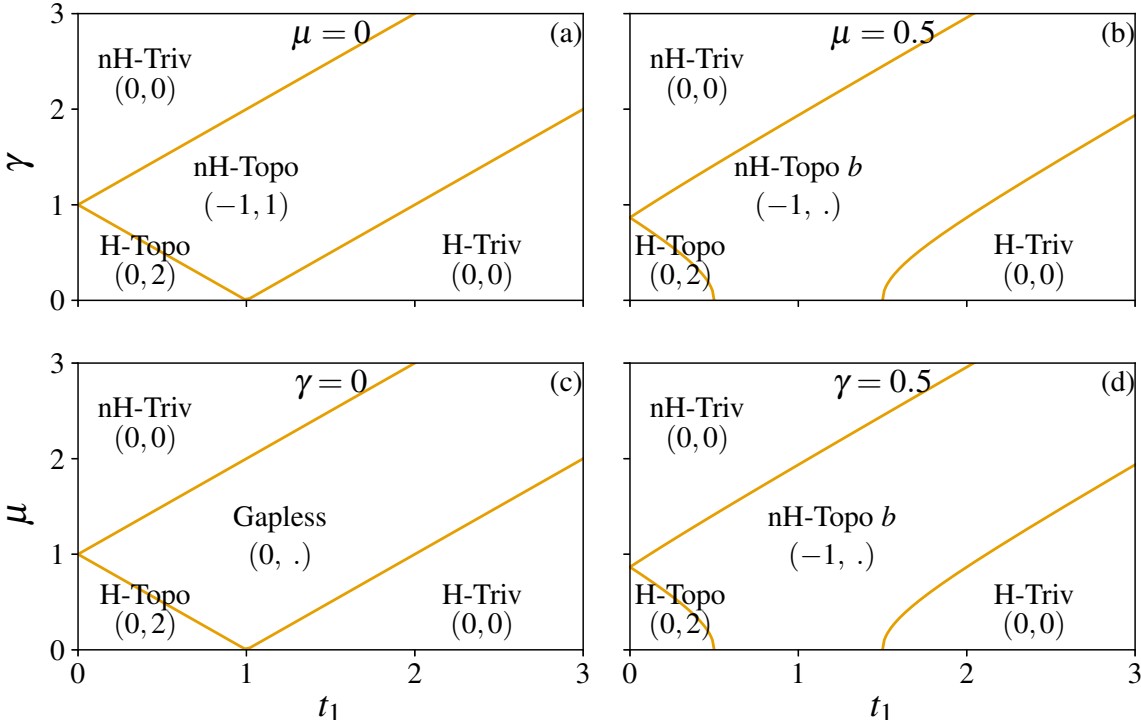

Figure 2: Phase diagram of the extended non-Hermitian SSH model as a function of $t_1$ and $\gamma$ for (a) $\mu = 0$, (b) $\mu = 0.5$, and as a function of $t_1$ and $\mu$ for (c) $\gamma = 0$ and (d) $\gamma = 0.5$. The different phases are labeled by the topological invariants $(\nu_+, \nu_-)$. While $\nu_-$ is quantized only for $\mu = 0$, it also acts as a good order parameter in the specific model we consider even when $\mu \neq 0$. (.) marks continuously varying values of $\nu_-$ in the phase. The phases "nH-Topo" and "nH-Topo $b$" are connected without gap closing, the label discriminate between the absence and presence of symmetries, and thus the quantization of $\nu_-$.

Hermitian trivial phase, while "nH-Triv" is connected to a trivial anti-Hermitian limit.

In the pseudo-hermitian limit $\gamma = 0$ [61–63], the system is pseudo-time-reversal $P_+$ symmetric with $u_p = \mathrm{Id}$, particle-hole $T_-$ symmetric with $u_t = \sigma^z$ and pseudo-hermitian $PH_-$ symmetric. The system now falls into the non-Hermitian class BDI$^\dagger$ [43] (group 14 in Ref. [42]), which is trivial following point-gap classification, but has the $\mathbb{Z}$ topological invariant $\nu_-$ for a real line gap. In this limit, the OBC and PBC phase diagrams coincide. "H-Topo" now admits purely imaginary edge modes, which are topologically stable (using the line gap criterium) and that partially survive in the gapless phase "Gapless" [64, 81].

Finally, when both $\gamma$ and $\mu$ are non-zero, the system is only particle-hole symmetric. It then falls into class $D^\dagger$ (group 34 in Ref. [42]) which admits $\nu_+$ as a $\mathbb{Z}$ topological invariant following the point gap classification, and $\nu_-/2 \mod 2$ as a $\mathbb{Z}_2$ topological invariant following the line gap classification. The "nH-Topo $b$" phase, i.e., the extension of "nH-Topo" to non-zero $\mu$, is non-trivial according to $\nu_+$. The "H-Topo" phase has non-trivial $\nu_-$. It is also characterized by non-separable energy bands surrounding $E = 0$.

Finally, we introduce the real space formulation of the previous topological winding numbers: [82–85]

$$\nu_+ = \text{AvTr}_{l:L-l}\left[Q^\dagger(QX - XQ)\right], \tag{40}$$

$$\nu_- = \text{AvTr}_{l:L-l}\left[\sigma^z Q^\dagger(QX - XQ)\right], \tag{41}$$

where $Q$ is the singular flattened Hamiltonian [57] (similar to Eq. (39) but in real space) and $X$ is the position operator. $\text{AvTr}_{l:L-l}$ means that we compute the average of the diagonal elements between sites $l$ and $L - l$. Note that these two formulations are subject to finite-size effects, caused by the presence of boundaries, and as such are not perfectly quantized in numerical computations. We generally take $l$ to be $L/4$ to limit these boundary effects.

# 6 Low-energy entanglement spectrum in the periodic chain

In this Section, we explore the properties of the entanglement spectra defined in Section 2.2 in the different phases of the extended SSH chain. In particular, we want to exemplify how the choice of either the biorthogonal or right reduced density matrix gives different insights into the topological properties of the Hamiltonian and the chosen many-body state. We consider a periodic system, and work with different many-body states at half-filling, depending on the structure of the energy bands in the complex plane. We compute both the eigenvalues and the singular values of the biorthogonal entanglement Hamiltonian, and compare them to the corresponding open Hamiltonian. While the open Hamiltonian can also present edge eigenstates, the conventional bulk-boundary correspondence holds for the singular value decomposition [40, 42, 43, 57] We only study the eigenvalues of the right entanglement matrices as they coincide with singular values in Hermitian matrices.

Diagonalization of a non-Hermitian Hamiltonian presents significant numerical noise, whose bound increases exponentially with the matrix size. In this paper, we present data from relatively small subsystems of 40 unit-cells for clarity. We performed a scaling analysis including subsystems of up to 100 unit-cells to confirm our results.

## 6.1 Chiral symmetric limit $\mu = 0$

The different phases of the system are here characterized by the two $\mathbb{Z}$ topological invariants $\nu_+$ and $\nu_-$ in Eqs. (40) and (41). We investigate whether the entanglement Hamiltonians inherit the topological properties of their system Hamiltonian.

### 6.1.1 Biorthogonal density matrix

We focus first on the biorthogonal entanglement spectrum. The numerical results are summarized in Fig. 3. We use the inverse participation ratio (IPR) to visualize the spatial extension of the eigenstates. It is a measure of the support of the eigenmodes: a state perfectly localized to a single site of the lattice will have an IPR of 1, while a state fully delocalized on all unit-cells and both sublattices will have an IPR of $2L$. The exact definitions employed are given in App. B.

In the phases "H-Topo" and "H-Triv" of Fig. 2, the PBC energy bands form two disconnected ellipsoids separated by the imaginary axis as in Fig. 1(b). It is therefore natural to compute the entanglement spectrum at half-filling from the state $|\phi_R\rangle = \prod_n d_R^{\dagger s_n}|0\rangle$, with $s_n = \delta_{\text{Re}(E_n)<0}$. These two phases are adiabatically connected to the Hermitian phases, and this definition is compatible with their respective Hermitian limit. The entanglement Hamiltonian then also respects all three symmetries ($T_+$, $T_-$ and $Ch$), and the entanglement spectrum is represented in Fig. 3.

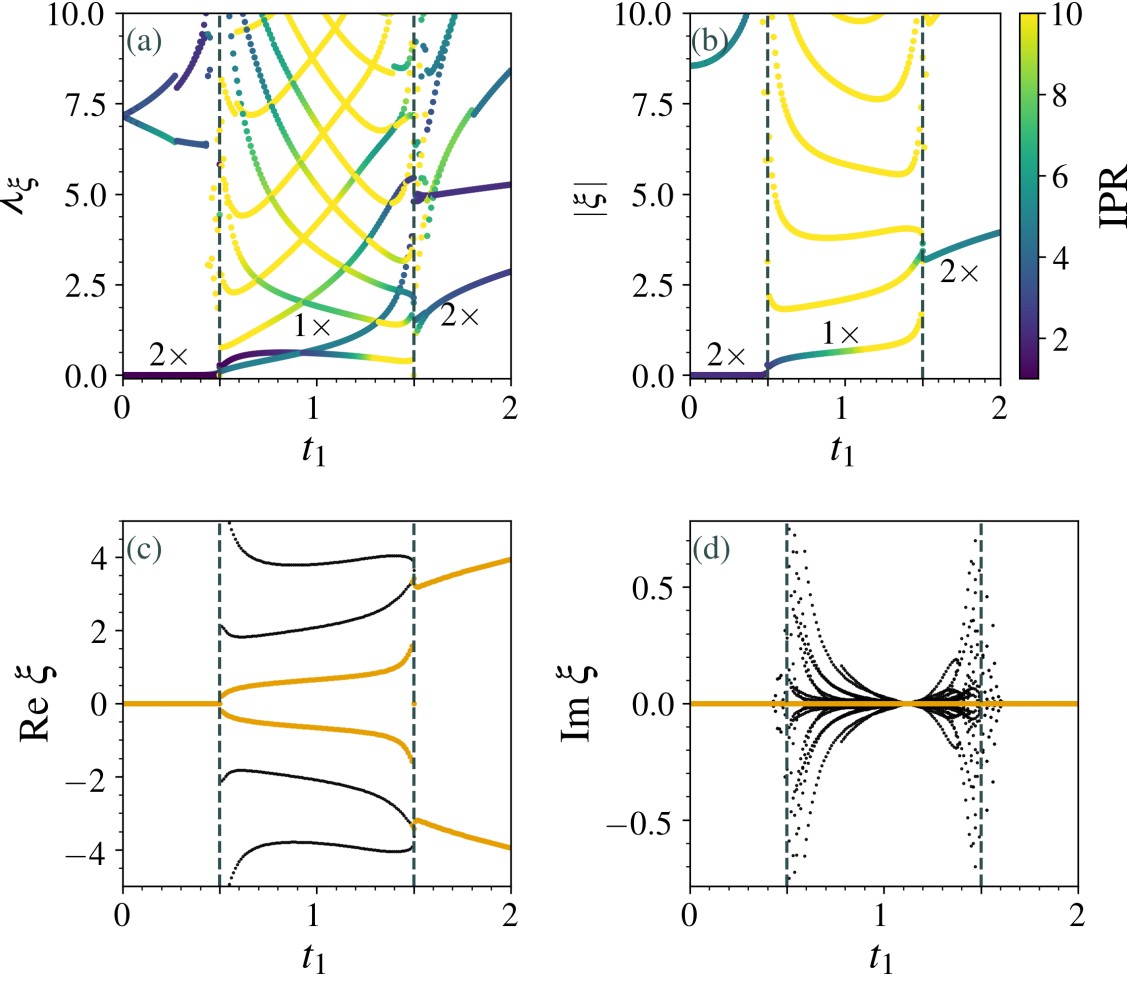

Figure 3: Singular values (a) and eigenvalues (b-d) of the biorthogonal entanglement Hamiltonian $H_E^{RL}$ as a function of $t_1$ for $\gamma = 0.5$ and $\mu = 0$. The total system is of length $L = 201$ and we consider a subsystem of size $l = 40$ unit-cells. In (a-b), colors represent the bilocalized inverse participation ratio of the corresponding singular modes $IPR^{SVD}$ and of the eigen modes $IPR^{RL}$ (see Eqs. (55) and (55) in App. B). In (c-d), we have highlighted (orange) the modes with the lowest real energies in absolute values. Phase transitions occur at $t_1 = 0.5$ and $t_1 = 1.5$ — marked by the dashed vertical gray lines—, characterized by an entanglement gap closing in the singular values and the presence of extended states at low-energy. We also indicate the degeneracy of the lowest-lying states. In the Hermitian topological phase, both the singular and eigen decompositions admit two zero modes which are localized at each end of the subsystem. In the non-Hermitian topological phase, we do not observe the zero singular mode that characterizes the open system.

The biorthogonal entanglement spectrum reveals the phase transitions occurring in the periodic system, and, despite being effectively open, shows a phase diagram matching the PBC one, when considering either eigen or singular values. "H-Topo" is characterized by the presence of two zero singular value modes, as expected from the OBC Hamiltonian. We also observe two corresponding zero energy modes in the whole phase. Each of these modes is localized at one end of the wire, up to finite-size effects, with the corresponding left- and right- eigenvectors exponentially localized on the same end. "H-Triv" is a trivial phase, and as

such, does not present any low entanglement energy excitation. We numerically compute the topological winding numbers from their real-space formula, and we show in Fig. 4 that, within numerical accuracy, the entanglement Hamiltonian indeed inherits the topological properties of the system Hamiltonian in these two phases .

In "nH-Triv", the PBC bands form two disconnected ellipsoids now separated by the real axis as in Fig. 1(c). This phase is in particular adiabatically connected to a purely anti-Hermitian trivial limit, which makes the more natural choice of occupation number in the many-body state to be $s_n = \delta_{\text{Im}(E_n)>0}$ if one wants to probe the topological property of the imaginary bands. Following the discussion in Section 4.2, this choice switches the roles of $T_+$ and $T_-$ symmetries, while conserving the chiral symmetry. The entanglement Hamiltonian satisfies

$$\sigma^z H_E^* \sigma^z = H_E \text{ and } H_E^* = -H_E. \tag{42}$$

The biorthogonal density matrix therefore still belongs to the same symmetry class. We observe no low energy or singular states and the topological invariants are zero. The modes with smallest absolute real part of the energy have an imaginary part close to $i\pi$ but have significant finite real part. For larger real parts, we expect a similar result, but we are limited by numerical accuracy and floating point precision.

Finally, in the phase "nH-Topo" the two bands are not separated but form a single ellipsoid encircling $E = 0$ as in Fig. 1(a). There is no longer any natural "ground state" allowing the study of a single band. We can either choose to select an arbitrary half-plane in energy space to populate, or to select states which can be smoothly deformed into each other. More precisely, choosing a mode $|R_{k_0,n}\rangle$ at momentum $k_0$, we select at $k = k_0 + \delta k$ the eigenstate $|R_{k,m}\rangle$ that maximizes $|\langle L_{k_0,n}|R_{k,m}\rangle|$. In practice, these two definitions coincide. Here we select the energy modes with negative real part, but similar results are obtained by using the negative imaginary ones. Our choice protects the chiral symmetry. The other symmetries would break in the thermodynamic limit due to the presence of purely imaginary modes. By taking $L$ odd (another possible choice is $L$ even and antiperiodic boundary conditions), we prevent the spontaneously breaking of the symmetries using finite-size effects, without affecting our results. We observe in this phase that the entanglement Hamiltonian breaks bulk-boundary correspondence: it has no zero singular value instead of the expected one. This is not a finite size effect, and is stable to perturbations. In fact, both real space topological invariants in Eqs. (40) and (41) are no longer quantized as the entanglement Hamiltonian becomes long range (approximately power-law decay of the hopping terms with strong oscillations, that saturate at a finite value independent of the subsystem size).

Such a breakdown of the bulk-boundary correspondence through the entanglement Hamiltonian is in sharp contrast with the ersatz of entanglement spectrum introduced in our own previous work [57]. This ersatz is based on the singular value decomposition of the single-particle Hamiltonian instead of a many-body eigenstate. The single-particle entanglement spectrum built from this SVD perfectly reproduces the physics of both the open and closed system.

### 6.1.2 Right density matrix

We now focus on the right density matrix and perform a similar analysis. Studying the left density matrix leads to the same results. Its entanglement spectrum is represented in Fig. 5. Only the time reversal symmetry $T_+$ is preserved — when it is also preserved in the biorthogonal case (in phase "nH-Triv", it is the new $T_-$ symmetry that is preserved). The entanglement Hamiltonian therefore belongs to the Hermitian AI class. The breakdown of the particle-hole symmetry can be understood from the following simple argument: The non-Hermitian term $\gamma$

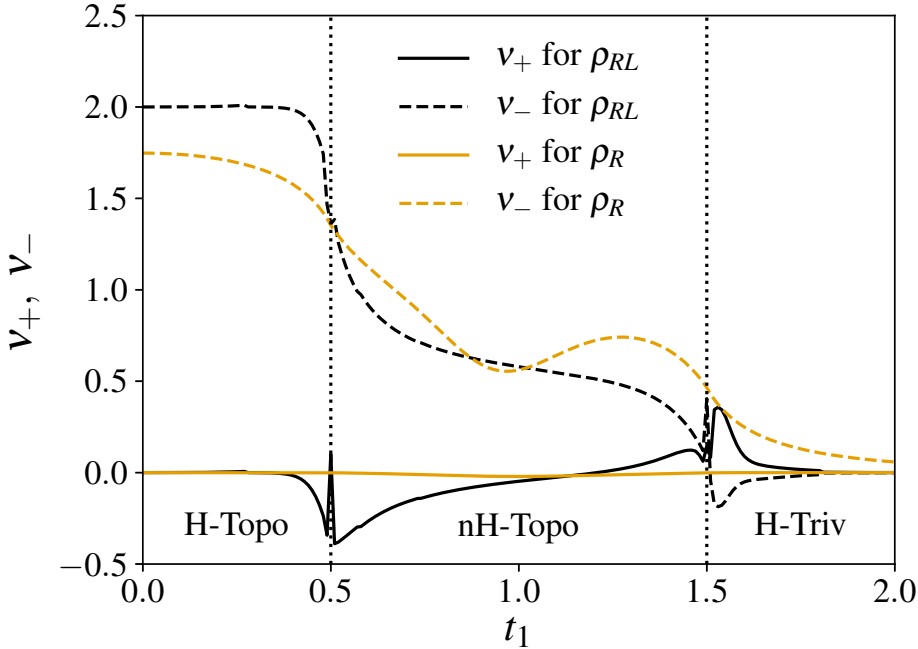

Figure 4: Topological invariants $\nu_+$ and $\nu_-$ of the biorthogonal and the right entanglement Hamiltonian in the chiral limit $\gamma = 0.5$ and $\mu = 0$, as a function of the hopping $t_1$. We consider a system of size $L = 801$ and a subsystem of size $l = 40$. The vertical dashed lines mark the PBC phase transitions. For the biorthogonal density matrix, in phase "H-Topo" and "H-Triv", the topological invariant takes the same values as in the original Hamiltonian. On the other hand, in the intermediate phase "nH-Topo", the real space topological invariant are no longer quantized as the Hamiltonian becomes long ranged. The phase transitions are nonetheless well marked. For the right density matrices, $\nu_+ = 0$ while $\nu_-$ is not quantized, as expected from a Hermitian Hamiltonian of class $AI$.

favors concentrating the wave function to the right of each unit cell. This means that $B$ sites tend to have larger occupancy number, hence breaking particle-hole and chiral symmetry. The class is trivial, and we do not observe any stable zero modes, whether in the singular or energy decomposition. In the "H-Topo" phase, the low singular or energy modes acquire a finite splitting in the presence of both $t_1$ and $\gamma$, though the low-energy modes stay localized on the boundaries. It can also be understood as a consequence of the larger occupancy of $B$ sites compared to $A$ sites. Note that this result means that line gap classification does not coincide with right density matrix classification. Indeed, the line-gap approach predicts a surviving $\mathbb{Z}$ classification, compatible with $\nu_-$, which is not observed here. The phase transitions are not characterized by a gap closing in the entanglement Hamiltonian. It is not just an effect of an ill-defined state in the intermediate phase. We performed a scaling analysis with respect to both $L$ and the length of the subsystem $\mathcal{A}$. Arbitrarily close to the transition in any line gapped phases, the entanglement Hamiltonian has a finite gap. Instead, the entanglement Hamiltonian transitions by becoming long-range.

## 6.2 Pseudo-Hermitian limit $\gamma = 0$

For $\gamma = 0$, the system falls into the class BDI$^\dagger$, which is trivial following point gap classification but with the $\mathbb{Z}$ topological invariant $\nu_-$ in the presence of a real line gap. The system with open-boundary conditions is argued to have topologically protected edge states with purely

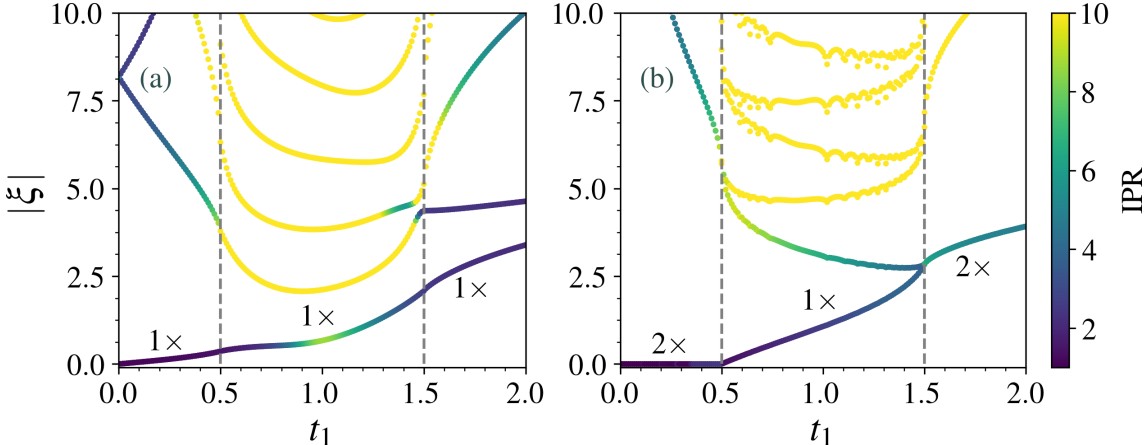

Figure 5: Absolute eigenvalues of the right entanglement Hamiltonian $H_E^R$ as a function of $t_1$ for (a) $\gamma = 0.5$, $\mu = 0$ and (b) $\gamma = 0$, $\mu = 0.5$. The total system is of length $L = 201$ and we consider a subsystem of size $l = 40$ unit-cells. Colors represent the inverse participation ratio of the corresponding singular and eigen modes and phase transitions are marked by the vertical dashed lines. We also indicate the degeneracy of the lowest eigenvalues. In the chiral limit, phase transitions are no longer visible and we observe no protected low-energy mode in the topological phase "H-Topo". In the pseudo-hermitian limit $\gamma = 0$, in phase "H-Topo", the right entanglement Hamiltonian has two zero-energy singular and energy entanglement modes protected by an emerging chiral symmetry. Interestingly, the gapless phase "Gapless" is gapped for the entanglement Hamiltonian. Separation between phases "Gapless" and "H-Triv" is not marked by a gap closing but by the coalescence of the lowest energy modes.

imaginary energies. We focus on the presence of such localized states directly in the entanglement spectrum.

### 6.2.1 Biorthogonal density matrix

Starting with the biorthogonal entanglement spectrum, we obtain similar results as in the previous section, as depicted in Fig. 6. We also observe here that the eigenvalues and singular values have a simultaneous change of behavior precisely where a phase transition occurs in the PBC system.

In "H-Topo", the energy spectrum of the system Hamiltonian is fully real and gapped, forming two separable bands with a real line gap. We select the state where all negative energy modes are occupied, by analogy with the Hermitian limit. This choice preserves the three symmetries $P_+$, $T_-$ and $PH_-$. The entanglement Hamiltonian is trivial according to the point gap classification of Refs. [42, 43]. As such, the singular and energy spectra of the entanglement Hamiltonian have no zero modes. Nonetheless, the BDI$^\dagger$ class admits the $\mathbb{Z}$ topological invariant $\nu_-$ following line gap classification. As shown in Fig. 7, $\nu_-$ is also quantized in the entanglement spectrum. Correspondingly, the singular spectrum admits two well separated low modes which correspond to two eigenmodes with purely imaginary energies. These two modes are exponentially localized at each edge of the subsystem, and match the corresponding edge modes observed in the OBC system.

When increasing $t_1$, we observe the transition to the gapless phase "Gapless". The spectrum of the PBC Hamiltonian now forms a cross on the real and imaginary axes. Selecting the many-body state following the deformation argument described in Section 6.1.1, we take $s_n = 1$ if

$E_n$ is real negative or imaginary positive. This indeed allows us to select one state at each momentum, and while it breaks both $T_-$ and $PH_-$ symmetries, it preserves the pseudo-time reversal symmetry. Note that $PH_-$ cannot be recovered in any many-body eigenstate: the imaginary modes cannot be avoided using finite-size effects and it is then not possible to satisfy the relation $s_n + s^*_{-n^*} = 1$ (in the absence of degeneracies in the spectrum). The entanglement Hamiltonian then falls into the trivial class AI$^\dagger$ (group 6). It is gapless, with extended eigen and singular modes. While in the OBC Hamiltonian the localized edge states survive in the gapless phase, they are not present in the entanglement Hamiltonian, indicating their more fragile nature as the edge modes can interact through the extended gapless modes. In the trivial phase "nH-Triv", the spectrum is again gapped and fully real, and we select the state with all negative modes occupied, respecting all symmetries. The entanglement Hamiltonian is correspondingly gapped, without low energy modes.

Finally, in the anti-Hermitian phase "nH-Triv", the energy spectrum is purely imaginary and we select states with negative imaginary parts. As discussed in Section 4.2, it transforms the symmetries $T_-$ and $PH_-$ into $T_+$ and $PH_+$ such that the entanglement Hamiltonian now verifies:

$$\sigma^z H_E^* \sigma^z = H_E \text{ and } \sigma^z H_E^\dagger \sigma^z = H_E. \tag{43}$$

It does not change the symmetry classification of the entanglement Hamiltonian and we observe no stable low singular or energy modes.

### 6.2.2 Right density matrix

We turn now to the right entanglement Hamiltonian. Similar to the previous limit, some symmetries are always spontaneously broken by our choice of states. As discussed in Section 4.3, the pseudo-Hermitian symmetry of the Hamiltonian leads to an emergent chiral symmetry of the right density matrix in phases "H-Topo" and "H-Triv". The entanglement Hamiltonian then falls into the AIII Hermitian class, which is topologically non-trivial, with $\nu_-$ the corresponding topological invariant. In the "H-Topo" region, we observe two exact zero modes localized at each side of the subsystem, shown in Fig. 5 and $\nu_-$ is quantized to 2, as shown in Fig. 7. The entanglement Hamiltonian is consequently topologically non-trivial. This means that the eigenvectors of the PBC Hamiltonian have a doubly degenerate Schmidt decomposition even though the Hamiltonian is trivial following the point-gap classification. The emergent symmetry also explains the quantization and stability of the right or left Berry phase observed in this limit in the periodic Hamiltonian [63, 79, 80]. In the "Gapless" phase, the initial density matrix and the entanglement Hamiltonian break all symmetries and are therefore trivial. The entanglement Hamiltonian is nonetheless gapped while the original Hamiltonian is gapless, with low but finite eigen modes power-law localized at each extremities of the subsystem, and higher-energy extended states. Finally, in "H-Triv", the chiral symmetry is restored, but the entanglement Hamiltonian is trivial.

### 6.3 Generic model

When both $\mu$ and $\gamma$ are non-zero, only the $T_-$ symmetry survives. The system then falls into the class $D^\dagger$ [43] (group 34 [42]), which admits the $\mathbb{Z}$ topological invariant $\nu_+$ following the point gap classification and the $\mathbb{Z}_2$ topological invariant $\nu_-/2$ mod 2 in a presence of a real line gap. The features of the entanglement spectrum and the state selection are then straightforwardly inherited from the two previous limits. In the "H-Topo", "H-Triv" and "nH-Triv" phases, the spectrum is line-gapped leading to a natural choice for the many-body state. Results are shown in Fig. 8. As in the two previous limits, singular values and eigenvalues show similar behavior and transition when the periodic system also goes through a critical point. For

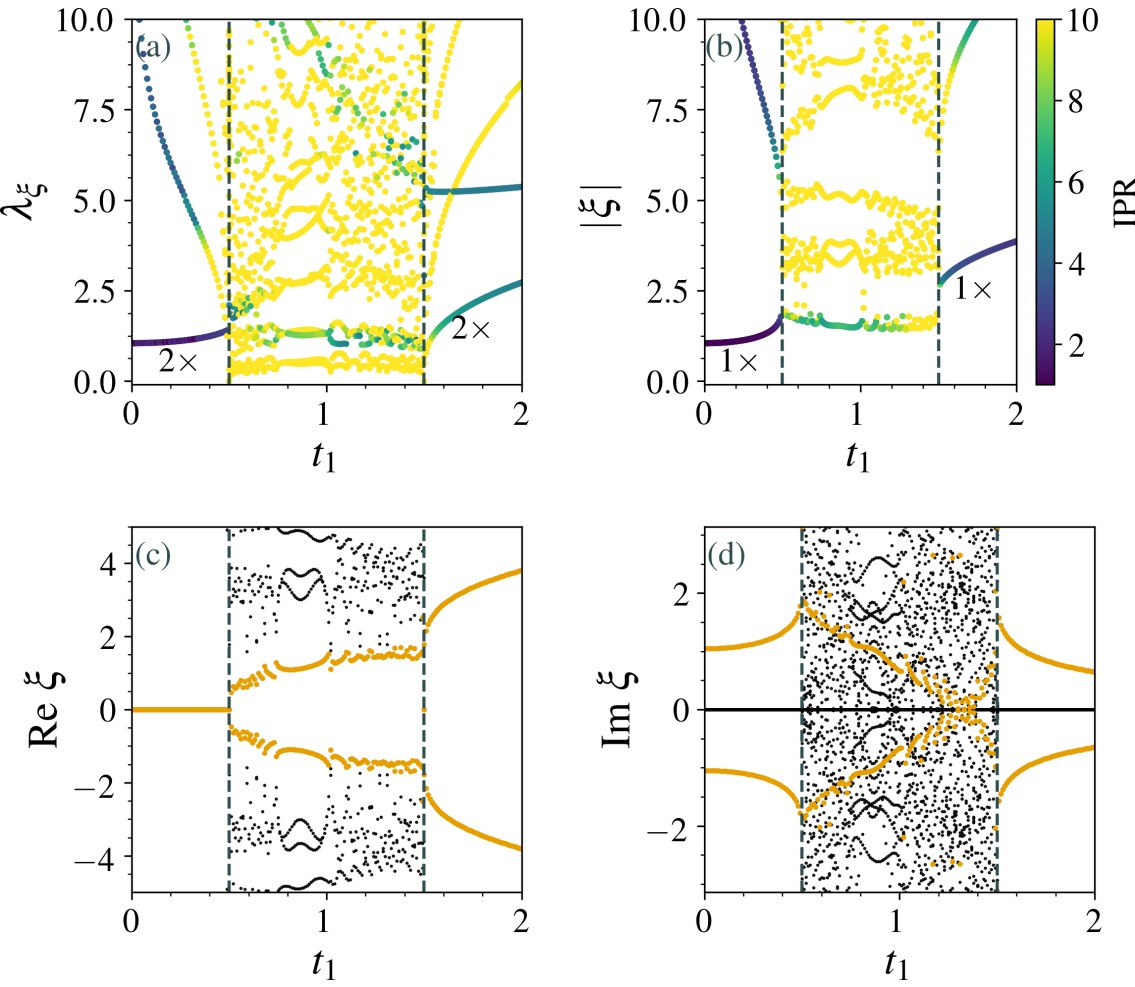

Figure 6: Singular values (a) and eigenvalues (b-d) of the biorthogonal entanglement Hamiltonian $H_E^{RL}$ as a function of $t_1$ for $\gamma = 0$ and $\mu = 0.5$. The total system is of length $L = 201$ and we consider a subsystem of size $l = 40$ unit-cells. In (a-b), colors represent the bilocalized inverse participation ratio of the corresponding singular modes $IPR^{SVD}$ and of the eigen modes $IPR^{RL}$ (see Eqs. (55) and (55) in App. B). The two gapped phases "H-Topo" ($t_1 < \frac{1}{2}$) and "H-Triv" ($t_1 > \frac{3}{2}$) are separated by the gapless phase "Gapless". The biorthogonal entanglement Hamiltonian presents a similar phase diagram. The noise in entanglement values is characteristic of finite size-effects in gapless phases. In (c) and (d), we highlight the eigenvalues with lowest absolute real part. In the "H-Topo" phase, we observe purely imaginary eigenstates exponentially localized at each extremity of the subsystem. The phase transitions are marked by the dashed vertical lines, and occur simultaneously for singular values and eigenvalues, at the parameter values predicted by the periodic system. While the corresponding edge states survive in the gapless phase for the OBC system, this is not the case for the entanglement Hamiltonian.

the biorthogonal entanglement spectrum, the "H-Topo" phase is characterized by the presence of modes with purely imaginary modes of the energy which are exponentially localized at the boundaries of the entanglement Hamiltonian (here localized), as in the open system (though phase boundaries do match the PBC phase diagram). The entanglement Hamiltonian correspondingly has non-trivial $\nu_-$. On the other hand, the right density matrix does not present

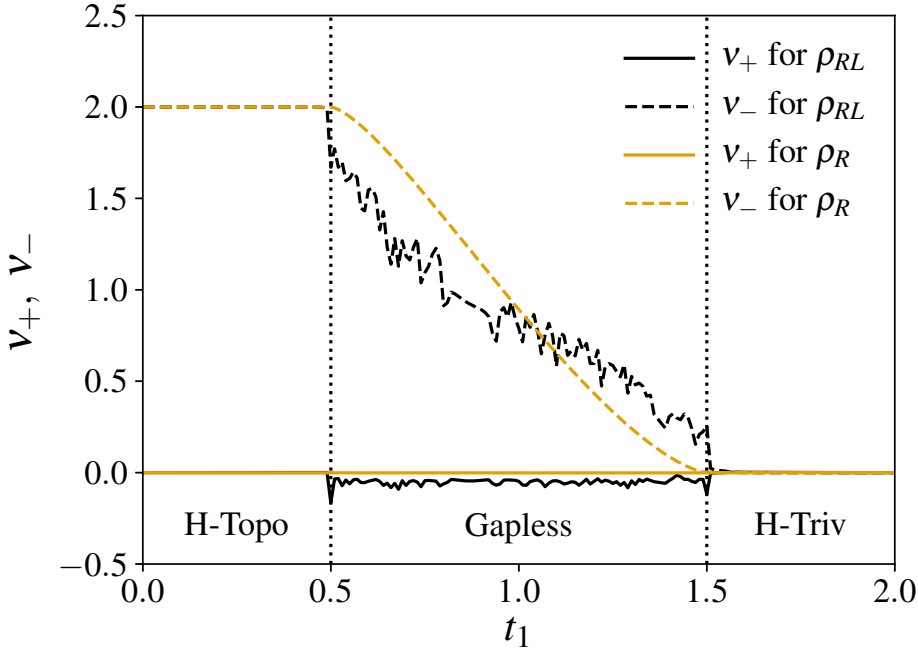

Figure 7: Topological invariants $\nu_+$ and $\nu_-$ of the biorthogonal and the right entanglement Hamiltonian in the pseudo-Hermitian limit $\gamma = 0$ and $\mu = 0.5$, as a function of the hopping $t_1$. We consider a system of size $L = 401$ and a subsystem of size $l = 40$. The vertical dashed lines mark the PBC phase transitions. $\nu_-$ is a good topological invariant for both the biorthogonal density matrix $\rho^{RL}$ and the right density matrix $\rho^R$ in the two line gapped phases "H-Topo" and "H-Triv". The results for $\rho^{RL}$ and $\rho^R$ exactly match in these two regions.

any stable low-energy mode. The $\gamma$ term, which preserves the chiral symmetry of the Hamiltonian breaks the chiral symmetry of the right density-matrix. The "H-Triv" and "nH-Triv" phases are topologically trivial and as such do not present any new features.

Finally, the "nH-Topo $b$" phase which is topologically non-trivial, has non-separable bands. As was the case in the previous examples, the entanglement spectrum then behaves differently from the system Hamiltonian. The entanglement Hamiltonian is long-range, with a non-quantized $\nu_+$, using the real space formula.

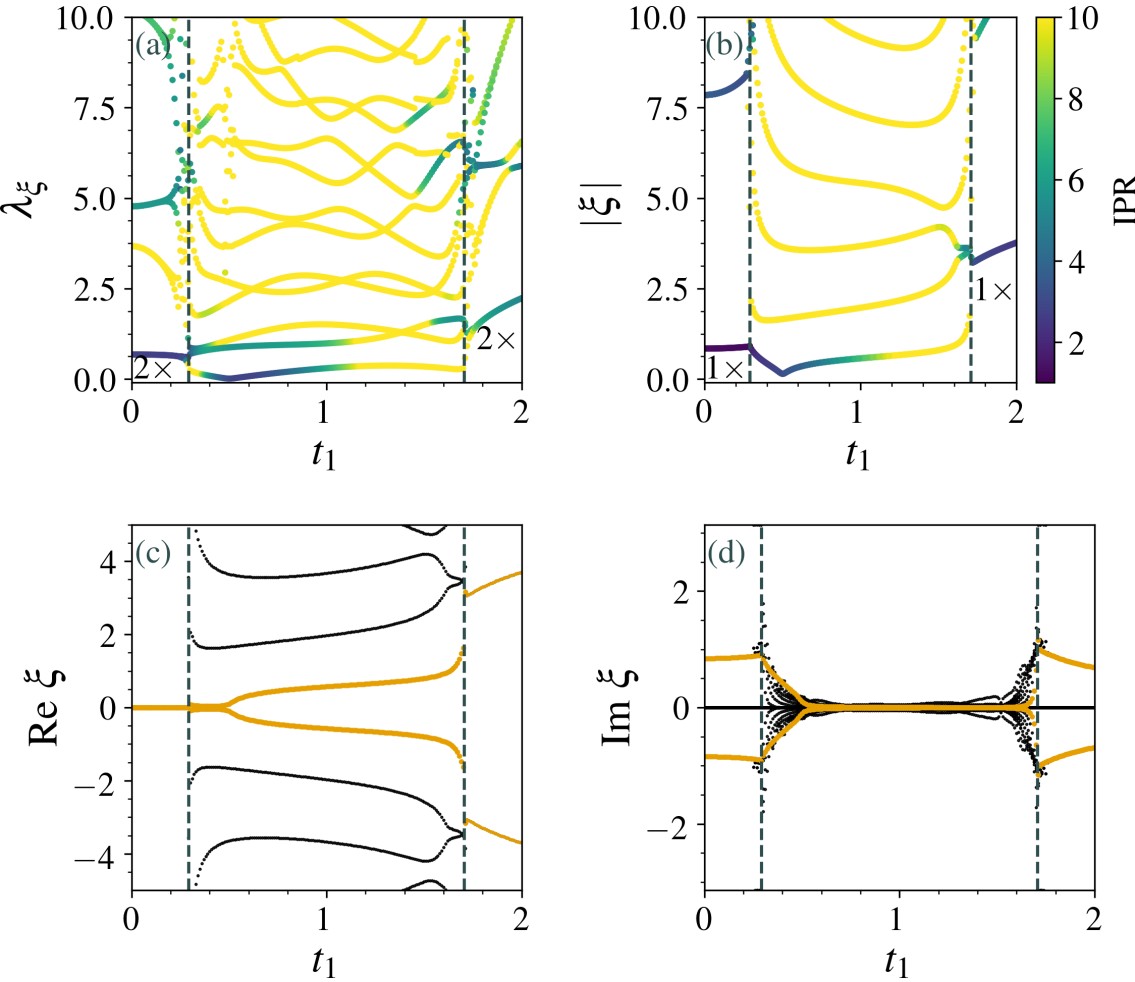

Figure 8: Singular values (a) and eigenvalues (b-d) of the biorthogonal entanglement Hamiltonian $H_E^{RL}$ as a function of $t_1$ for $\gamma = \mu = 0.5$. The total system is of length $L = 201$ and we consider a subsystem of size $l = 40$ unit-cells. In (a-b), colors represent the bilocalized inverse participation ratio of the corresponding singular modes $IPR^{SVD}$ and of the eigen modes $IPR^{RL}$ (see Eqs. (55) and (55) in App. B). In (c) and (d), we highlight the eigenvalues with lowest absolute real part. In the phase "H-Topo", we observe two localized edge states with purely imaginary energies. These states are nonetheless not topologically stable. The intermediate phase "nH-Topo" has non-separable energy bands, which leads to a non-local entanglement Hamiltonian and delocalized modes. The phase transitions occurs at $t_1 = 1 \pm \frac{1}{\sqrt{2}}$, as predicted by the PBC Hamiltonian, and are marked by the dashed vertical lines. Singular values and eigenvalues transition simultaneously.

# 7 Two-dimensional models: from Chern insulators to non-Hermitian topology

In this Section, we compute the entanglement spectrum of several two-dimensional non-Hermitian topological models in order to illustrate the properties and limits of our approach. Using three different models, we study the two entanglement spectra, obtained from $\rho^R$ and $\rho^{RL}$, in different topological phases and discuss when they give insight on the properties of the system Hamiltonian. In all the following examples, the Hamiltonian is defined on a two-

dimensional torus with periodic boundary conditions. The subsystem we use to define the entanglement spectrum is a cylinder, periodic in the $x$-direction, but finite in the $y$-direction. In simulations, we take systems with $100 \times 100$ unit cells, and the cylinder has a length of 40 unit-cells. This cylinder geometry is also what we denote by open boundary conditions in this section.

## 7.1 Non-Hermitian Chern insulator

We start by studying the generic non-Hermitian extension of a Chern insulator introduced in Refs. [86,87]. Its Bloch Hamiltonian reads

$$\mathcal{H}_{\text{Chern}} = \sum_{\vec{k}} (c^{\dagger}_{\vec{k},\uparrow}, c^{\dagger}_{\vec{k},\downarrow}) \left[\vec{n}(\vec{k}) + i\vec{d}(\vec{k})\right] \cdot \vec{\sigma} \ (c_{\vec{k},\uparrow}, c_{\vec{k},\downarrow})^{T}, \tag{44}$$

with $\vec{\sigma} = (Id, \sigma^{x}, \sigma^{y}, \sigma^{z})$ the vector of Pauli matrices, $c^{\dagger}_{\vec{k},\alpha}$ the fermionic creation operator at momentum $\vec{k}$ with spin $\alpha = \uparrow, \downarrow$ and

$$\vec{n}(\vec{k}) = (0, \Delta_{x} \sin k_{x}, \Delta_{y} \sin k_{y}, -\mu - t \cos k_{x} - t \cos k_{y}), \tag{45}$$
$$\vec{d}(\vec{k}) = (0, \gamma_{x}, \gamma_{y}, \delta\mu). \tag{46}$$

Here $\mu$ corresponds to a Zeeman field, $t$ a hopping between lattice sites, $\Delta_{x}$ and $\Delta_{y}$ are spin orbit couplings, and $\gamma_{x}$ and $\gamma_{y}$ are constant dissipative spin-flip terms, while $\delta\mu$ is a local source or drain coupled to the spin polarization. In the following, for simplicity, we take $t = \Delta_{x} = \Delta_{y} = 1$. In the Hermitian limit $\vec{d}(\vec{k}) = \vec{0}$, the system is topologically non-trivial for $|\mu| < 2t$. Two topological phases with opposite Chern number $\pm 1$ are separated by a gapless line at $\mu = 0$. These two phases are characterized by the presence of chiral edge-modes when considering open boundary conditions. Similar structures are observed in the entanglement spectrum [73, 82, 88, 89]. When $\mu > 2t$, the system becomes trivial. The topological phases are not protected by any symmetry, though the Hermitian model is particle-hole symmetric.

When all parameters are non-zero, the system has no special symmetries and falls into class $A$ ($D^{\dagger}$ if $\delta\mu = 0$), which is topologically trivial following point-gap classification, but admits a $\mathbb{Z}$ topological invariant following the line-gap classification [43]. This topological invariant is nothing but the Chern number, and the corresponding phases are the extension of the Hermitian phases. In this section, we therefore limit ourselves to this extension, i.e., we introduce non-Hermitian terms without breaking the line gap (and hence the point gap). Due to this line gap, the eigenvalues are well separated into two different energy bands. When we consider a cylinder geometry, the system still admits one localized chiral edge-mode at each edge. The two modes have opposite chirality, and one is amplified while the other is dissipated.

The system presents a real line gap as shown in Fig. 9(a). We therefore select the many-body state at half-filling where the levels with negative real part are occupied, and compute the entanglement spectrum over a cylinder periodic in the $x$ direction. In the topological phases, the biorthogonal entanglement spectrum presents chiral edge modes as shown in Fig. 9(c-d), and the entanglement spectrum has the same Chern number as its system Hamiltonian. The edge modes are dissipative, with finite imaginary part, similarly to the original Hamiltonian with open-boundary conditions. The chirality of the amplified and dissipated modes are the same in the entanglement Hamiltonian $H^{RL}_{E}$ and the system Hamiltonian. The right entanglement Hamiltonian—whose spectrum is shown in Fig. 9(b)—also falls into class $A$, and has similar topological properties with the same Chern number as the initial Hamiltonian. In the trivial phase, the entanglement Hamiltonians do not have any special feature. Transitions occur as predicted by the PBC Hamiltonian.

In this model, the entanglement spectrum is therefore able to correctly predict the properties of the line-gapped topological phases.

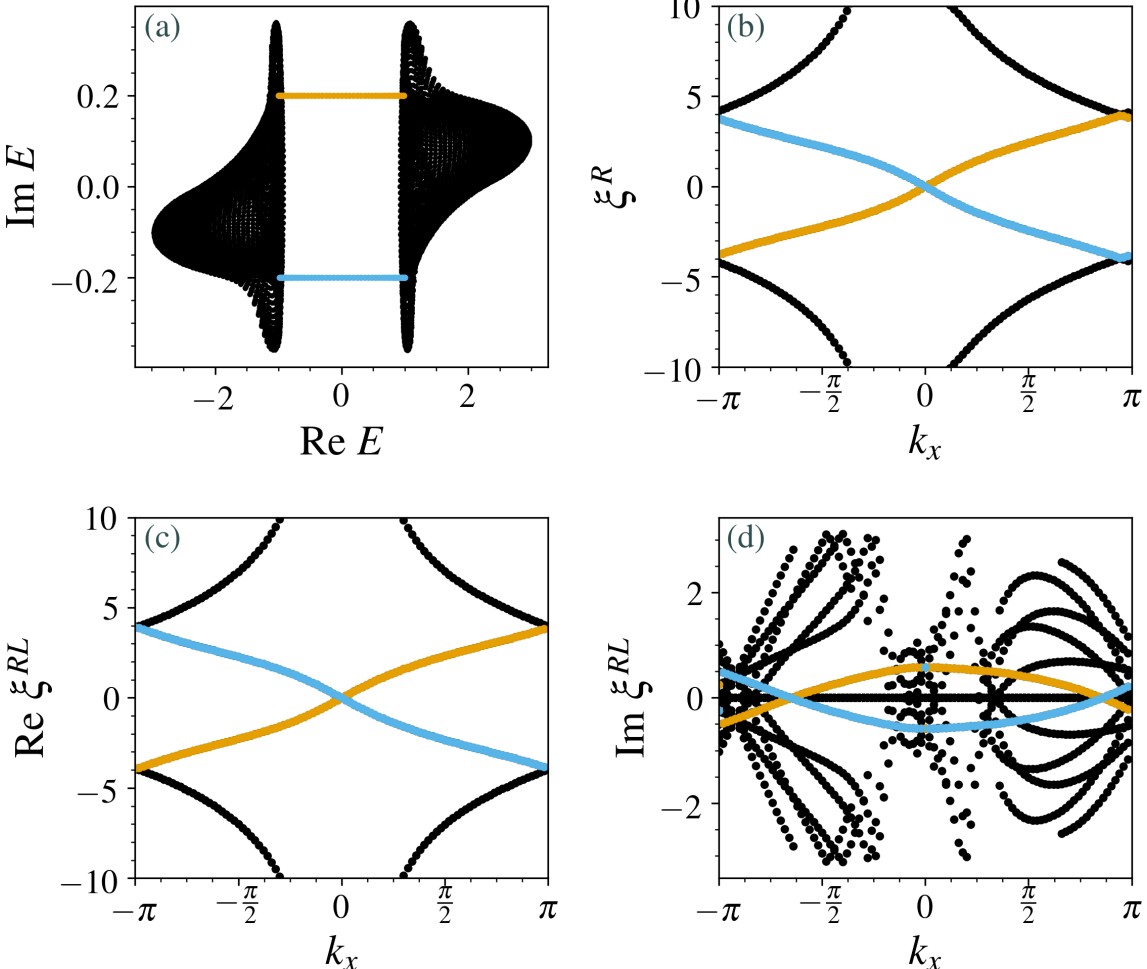

Figure 9: (a) Energy spectrum of the non-Hermitian Chern insulator for $\gamma_x = 0.2$, $\gamma_y = 0.3$, $\delta\mu = 0.1$ and $\mu = 1$, deep in the non-Hermitian topological phase on a cylinder geometry. Two edge modes of opposite chirality with finite imaginary part are present. (b) Right entanglement spectrum obtained for the same parameters as a function of the conserved momentum $k_x$. Topological edge modes are also present. (c-d) Real and imaginary part of the biorthogonal entanglement spectrum. The chiral edges have the same sign of the imaginary part as in the original Hamiltonian close to $k_x = 0$.

## 7.2 Non-Hermitian $\mathbb{Z}$ topological phase

We now turn to a simple model in class DIII$^\dagger$, whose Bloch Hamiltonian is parametrized by

$$\vec{n}(\vec{k}) = (0, \Delta_x \sin k_x, \Delta_y \sin k_y, 0), \tag{47}$$

$$\vec{d}(\vec{k}) = (\mu - t_x \cos k_x - t_y \cos k_y, 0, 0, \delta(\sin k_x + \sin k_y)), \tag{48}$$

using the notations of Eq. (44). $t_x$, $t_y$ are dissipative hopping terms, $\Delta_x$ and $\Delta_y$ are normal spin-orbit hoppings, $\mu$ is a spin-dependent source and drain and $\delta$ is a dissipative spin-orbit contribution. The model has a $T_-$ symmetry with $u_t = \sigma^x$, $P_+$ symmetry with $u_p = \sigma^y$ and a pseudo-Hermitian symmetry $PH_-$. It admits a $\mathbb{Z}$ topological invariant following point gap classification [42, 43]. DIII$^\dagger$ is also non-trivial in the line gap classification. We discuss an example in the following section. We fix $t_x = t_y = \Delta_x = \Delta_y = 1$. This model was briefly

discussed in Ref. [42] in the limit $\delta = 0$. Then, for $|\mu| < 2$, the Hamiltonian is topologically non-trivial. The two-bands are not separable as shown in Fig. 10(a). and the OBC Hamiltonian admits two degenerate singular zero modes, while nonetheless it has no edge modes in the energy spectrum, as shown in Fig. 10(b-c).

We compute the entanglement spectrum in the topological phase. We select the many-body state where all states with negative real energy are selected, to preserve the $T_-$ symmetry. The $PH_-$ symmetry can also be preserved by considering an antiperiodic torus, though this choice does not significantly affect the obtained entanglement spectra. In the following, we only show the entanglement spectrum computing the many-body state of the more conventional periodic torus geometry. In the limit $\delta = 0$, the non-Hermitian terms are diagonal in momentum space and $\rho^{RL}$ and $\rho^R$ coincide as the many-body state is the ground state of a gapless Dirac Hermitian Hamiltonian. It has four Dirac cones at the protected momenta $\vec{k} = (0,0)$, $(0,\pi)$, $(\pi,0)$ and $(\pi,\pi)$. The entanglement spectrum of such a many-body state does not present any stable zero modes, though it still supports some low-energy gapped modes due to the presence of the two sets of two Dirac cones with opposite chirality. For small non-zero $\delta$, in the topological phase, this picture is still valid, as shown in Fig. 10(d-f).

## 7.3 Non-Hermitian pseudo-Hermitian $\mathbb{Z}_2$ insulator

Finally, we introduce a non-Hermitian extension of a $\mathbb{Z}_2$ insulator in the same DIII$^\dagger$ class. We now focus on line gap classification and show that the topological properties of the two entanglement Hamiltonians can differ due to the presence of an emergent chiral symmetry in the right entanglement Hamiltonian. The class admits a $\mathbb{Z}_2$ topological invariant in the presence of a real line gap [43], which can be expressed as an extension of the Kane-Mele invariant [1]. By analogy with the Hermitian DIII class, we consider a model with four bands. The toy Hamiltonian reads

$$H_{KM} = \Delta_x \sin k_x \sigma^{xx} + \Delta_y \sin k_y \sigma^{xy} + (\mu - 2t_x \cos k_x - 2t_y \cos k_y)\sigma^{y0} + i\gamma\sigma^{zz}, \qquad (49)$$

where $\sigma^{\alpha\beta} = \sigma^\alpha \otimes \sigma^\beta$, $\alpha, \beta = x, y, z, 0$. The system is $T_-$ symmetric with $u_t = \sigma^{yy}$, $P_+$ symmetric with $u_p = \sigma^{xy}$ and $PH_-$ symmetric with $u_{ph} = \sigma^{z0}$. In the Hermitian limit $\gamma = 0$, it has been introduced in Ref. [90], and is topologically non-trivial for $|\mu| < 2|t_x| + 2|t_y|$. In a cylinder geometry, it presents two free chiral edge modes with opposite chirality at each edge. Introducing a small anti-Hermitian parameter $\gamma$ does not break the real line gap (Fig. 11(a)), and preserve the topological phases. Indeed, as shown in Fig. 11(b-c), both singular and eigen decompositions of the Hamiltonian still present similar zero edge modes.

Since this model has a real line gap, we compute the entanglement spectrum of the many-body state where all states with negative real part of the energy are occupied, in the topological phase. Results are shown in Fig. 11(d-f). The biorthogonal entanglement Hamiltonian presents the same edge states as the open model, both in its singular value decomposition and itś eigendecomposition. It therefore faithfully captures the topological properties of the initial Hamiltonian. On the other hand, the right entanglement Hamiltonian has gapped low-energy modes and is actually topologically trivial. Indeed, as discussed in Sec. 4.3, the pseudo-Hermitian symmetry of the non-Hermitian Hamiltonian transforms into a chiral symmetry for the right entanglement Hamiltonian. On the other hand, our choice of non-Hermitian perturbation prevents the $T_-$ and $P_+$ symmetry to carry over to $H_E^R$. $H_E^R$ then falls into the trivial Hermitian class $D$.

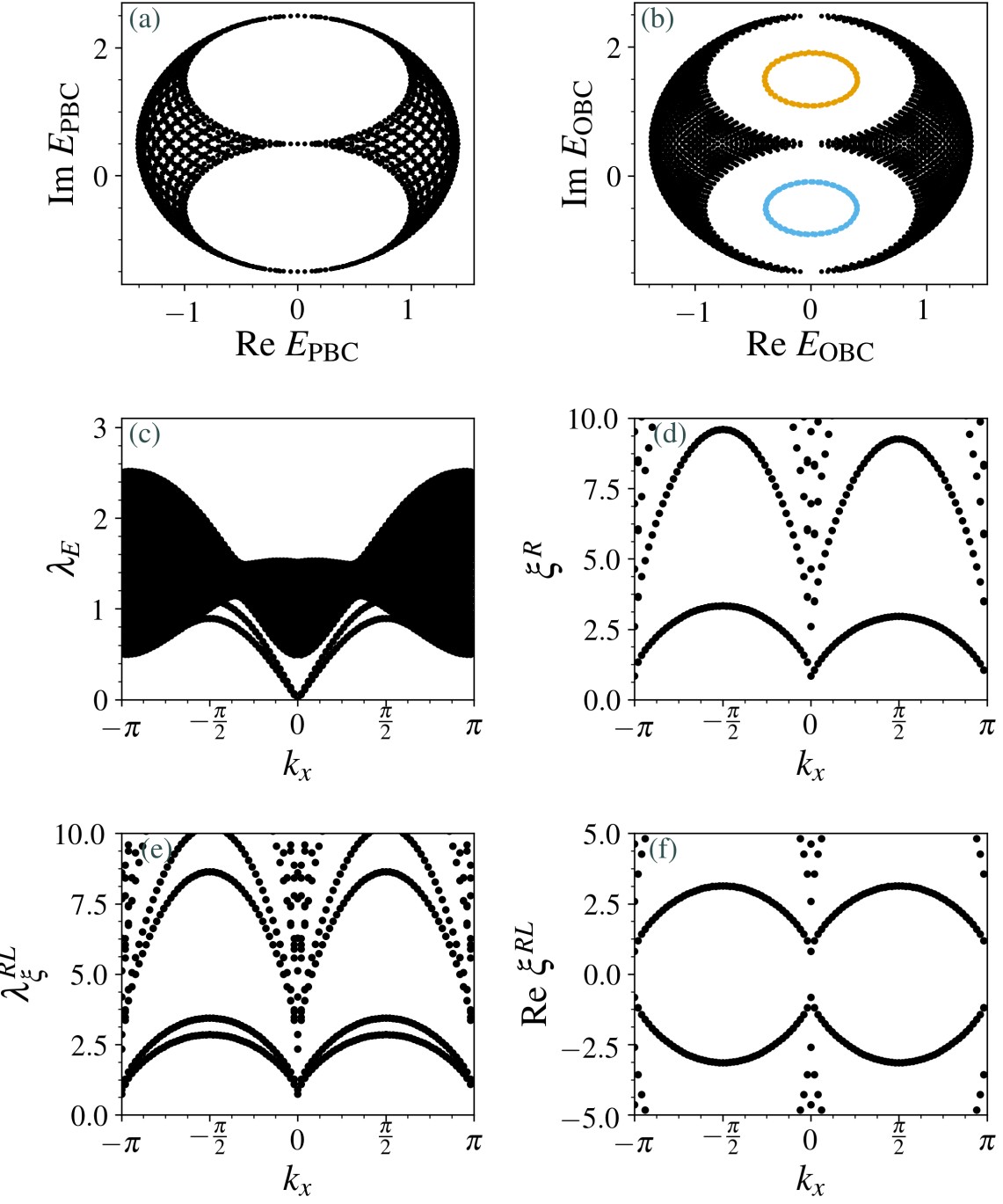

Figure 10: We fix $\mu = 0.5$ and $\delta = 0.1$. (a) Energy spectrum of the model in Eq. (48) for periodic boundary conditions. (b) Energy spectrum on a cylinder geometry. The highlighted bands are not edge states but two fixed momentum bands ($k_x = 0$ for the negative imaginary parts and $k_x = \pi$ for the positive imaginary parts). The discontinuity in the band structure is due to the instability of the eigenvalues in non-Hermitian systems [57]. (c) Singular value spectrum of the OBC model. Two degenerate zero singular modes appear and are topologically protected. (d) Entanglement spectrum of the right density matrix. (e) Singular values of the biorthogonal entanglement Hamiltonian. (f) Real part of the biorthogonal entanglement Hamiltonian. In (d-f), we observe low energy gapped edge modes which are caused by the presence of two set of Dirac cones with opposite chirality, but no stable zero modes as in the singular value decomposition of the Hamiltonian

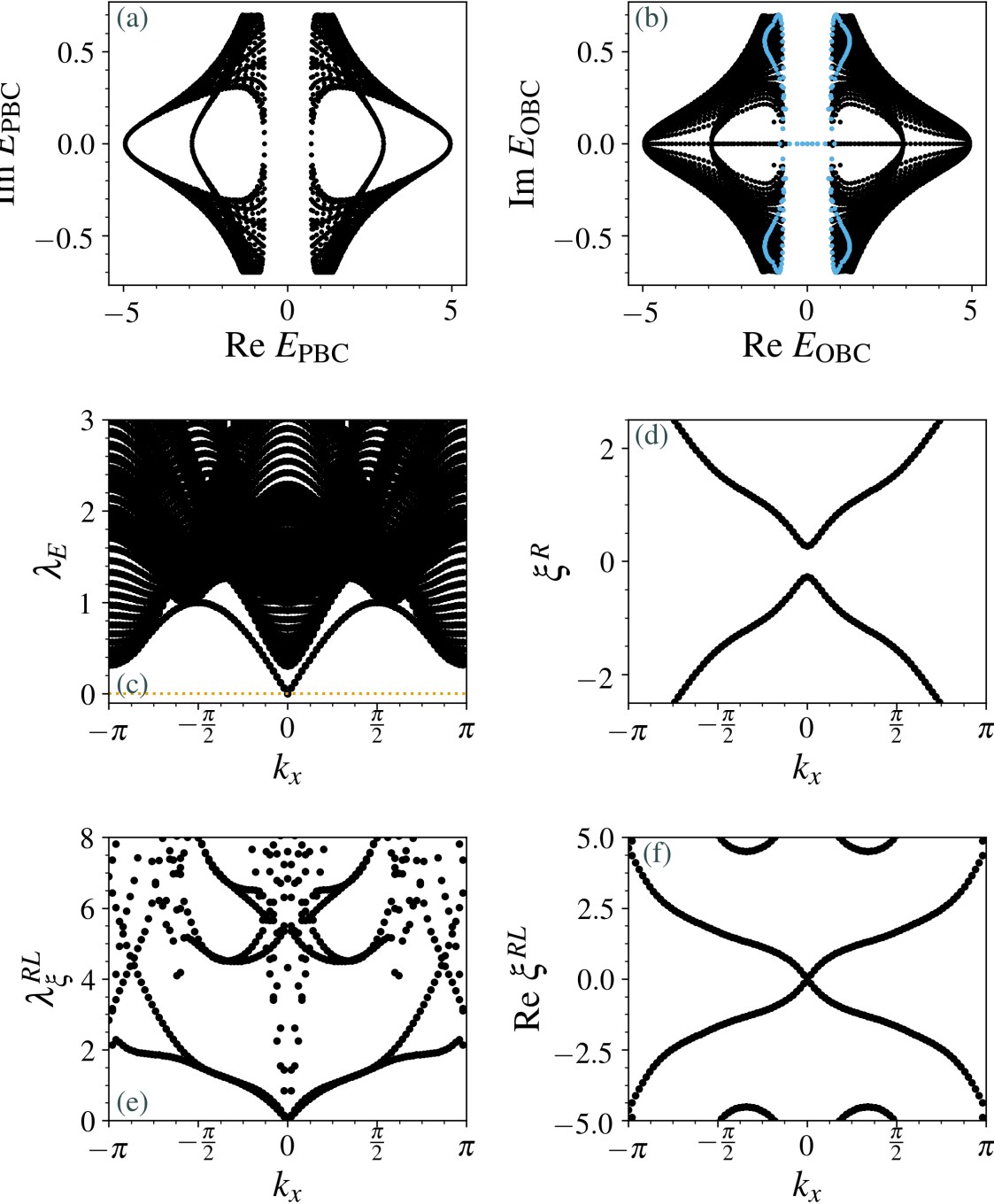

Figure 11: We fix $t_x = t_y = 2$, $\Delta_x = \Delta_y = \mu = 1$ and $\gamma = 0.7$. (a) Energy spectrum of the model in Eq. (49) for periodic boundary conditions. (b) Energy spectrum on a cylinder geometry. The highlighted bands are the states with the lowest real part of the energy at each momentum $k_x$. The low-energy edge modes are stuck on the real axis. (c) Singular value spectrum of the OBC model. Two chiral zero singular modes appear at each edge and are topologically protected. The dotted orange line is a guide to the eye. (d) Entanglement spectrum of the right density matrix: the edge modes are not protected and become gapped (e) Singular values of the biorthogonal entanglement Hamiltonian. (f) Real part of the biorthogonal entanglement Hamiltonian. In (e-f), we observe the same low energy edge modes as in the open Hamiltonian.

# 8   Conclusions and discussions

In this work, we have discussed the properties of the many-body density matrices and entanglement Hamiltonian in topological non-Hermitian systems. After discussing two possible definitions of density matrices, we have shown that both Wick's theorem and Peschel's formula are valid in non-interacting non-Hermitian settings, even for non-diagonalizable Hamiltonians. We have then studied how the symmetries of the Hamiltonian maps onto the density matrices and the entanglement Hamiltonian. As opposed to Hermitian models, the choice of a many-body state, like a filled band for insulator, is not always unambiguous. We propose to base this choice on symmetry. For the biorthogonal density matrix, depending on the choice of many-body state, different symmetries can be realized at fixed half-filling. For the right (or left) density matrix, most of the symmetries of the starting Hamiltonian do not naturally carry on to the entanglement Hamiltonian, contrarily to what happens in Hermitian system. Nonetheless, the pseudo-Hermitian symmetry $PH_-$ : $H = -u_{ph}H^\dagger u_{ph}^\dagger$ may lead to an emergent chiral symmetry which translates into topologically non-trivial right and left wave-functions.

To exemplify these different approaches, we have studied the entanglement Hamiltonian of several archetypal models in one and two dimensions. Starting from the periodic Hamiltonian, we have found that the biorthogonal entanglement spectrum inherits the topological properties of the initial Hamiltonian as long as the system has separable bands. The singular and edge modes present in the open Hamiltonian are present in the entanglement Hamiltonian, and the corresponding topological invariants carry on. On the other hand, the right entanglement spectrum does not reproduce all the features of the original Hamiltonian. As symmetries of the system Hamiltonian do not straightforwardly carry to the right entanglement Hamiltonian, the latter can present topological features in phases that are trivial following the point gap classification, or conversely be trivial in topological phases. For non-separable bands, both entanglement Hamiltonians fail to reproduce the characteristic topological properties of the original Hamiltonian, in contrast with the singular value approaches discussed in Ref. [57]. The singular zero-modes typically present in these phases are not present in the entanglement Hamiltonian, for all the many-body states we have considered. It appears then, that the bulk-boundary correspondence holds for the ES in line-gapped Hamiltonians, when considering the bi-orthogonal density matrix. We have indeed analytically shown that the entanglement Hamiltonian belongs to the same symmetry class as the original Hamiltonian, under symmetry dependent assumptions (e.g., the absence of purely imaginary eigenmodes) which are generically valid in line-gapped phases, and numerically shown that their corresponding topological invariants then match. This result can be seen as a consequence of the fact that line-gapped non-Hermitian Hamiltonians are smoothly deformable into (anti-)Hermitian ones [43], for which we know the bulk-boundary correspondence generally holds. The right density matrix carries information on the topological properties (degeneracies and zero modes in the entanglement spectrum, Chern number of the corresponding entanglement Hamiltonian...) of the many-body right eigenstates themselves. The subject of the classification of these matrices following from the topological properties of the system Hamiltonian can be relevant to experiments with post-selection.

The approach we develop in this paper is a first step towards the generalization of the non-Hermitian topological classifications to true many-body physics. Indeed, it is highly non-trivial to generalize the approaches introduced in Refs. [40, 42, 43], as the point gap classification relies on the singular value decomposition of the single-body Hamiltonian, which cannot be simply related to the eigen or singular decomposition of the many-body Hamiltonian. Asking the question whether the many-body states have topological properties, characterized by their entanglement spectrum, allows us to circumvent this difficulty.

Performing a similar analysis starting from an open system could further improve our un-

derstanding of the structure of these states. A complete study is left for future works due to more challenging numerics. Similarly, it would be interesting to generalize this approach to interacting systems [91], either through standard exact computation or through modified MPS algorithm, though the numerical instabilities inherent to non-Hermitian system may limit these approaches. In this paper, we considered non-interacting fermionic models because it allowed us to use Peschel's formula and study much larger systems. The rest of our approach should be directly applicable to interacting systems.

Following this work, a similar approach was developed in Ref. [92] to study the entanglement entropy and therefore the effective low-energy conformal field theory describing critical points in non-Hermitian models.

## Acknowledgements

We thank Thomáš Bzdušek, Adrien Bouhon, Maria Hermanns, Vardan Kaladzhyan, Flore Kunst and Simon Lieu for useful discussions.

**Funding information** This work was supported by the ERC Starting Grant No. 679722, and the Roland Gustafsson's Foundation for Theoretical Physics. N.R. was supported by the Department of Energy Grant No. DE-SC0016239, the National Science Foundation EAGER Grant No. DMR 1643312, Simons Investigator Grant No. 404513, ONR Grant No. N00014-14-1-0330, the Packard Foundation, the Schmidt Fund for Innovative Research, and a Guggenheim Fellowship from the John Simon Guggenheim Memorial Foundation.

## A Antecedent of non-Hermitian correlation matrices with Jordan blocks

In this Appendix, we show how to find the Gaussian antecedent of a correlation matrix that forms an arbitrary Jordan block of size $n$. Generalization to an arbitrary correlation matrix is straightforward. Each eigenspace (corresponding to independent Jordan blocks) can be treated separately.

We start by computing the correlation matrix obtained when the entanglement Hamiltonian is a single Jordan block of size $n$. Let $H_E = \sum_{j=1}^{n} \varepsilon c_{R,j}^{\dagger} c_{L,j} + \sum_{j=1}^{n-1} c_{R,j}^{\dagger} c_{L,j+1} = \vec{c}_R^{\dagger} J(\varepsilon) \vec{c}_L$ with $J(\varepsilon)$ the $n-$dimensional Jordan block with eigenvalue $\varepsilon$, in some arbitrary biorthogonal basis. Then the corresponding two-site correlation matrix $M$ defined by $M_{i,j} = \left\langle c_{R,j}^{\dagger} c_{L,i} \right\rangle$ is the banded matrix

$$M = \begin{pmatrix} m_1 & m_2 & \cdots & \cdots \\ 0 & m_1 & m_2 & \cdots \\ \vdots & \ddots & \ddots & \ddots \\ 0 & \cdots & 0 & m_1 \end{pmatrix}, \tag{50}$$

with $m_1 = \frac{e^{\varepsilon}}{1+e^{\varepsilon}}$ and $m_2 = -\frac{e^{\varepsilon}}{(1+e^{\varepsilon})^2}$ (the higher diagonals are generally non-zero, but they are not relevant to our discussion). As $m_2$ is non-zero, this matrix cannot be diagonalized and forms a single $n-$dimensional Jordan block. We denote by $Q$ the invertible matrix such that $M = Q J(m_1) Q^{-1}$.

We now prove that any correlation matrix forming a single Jordan block admits a Gaussian antecedent. Let $C$ be a correlation matrix, and $P$ an invertible matrix be such that

$C = PJ(s)P^{-1}$. Using

$$\text{Tr}\left(c_\alpha^\dagger c_\beta e^{-\vec{c}^\dagger H_E \vec{c}}\right) = \sum_{m,n} P_{\beta,m} \text{Tr}\left(f_{R,n}^\dagger f_{L,m} e^{-\vec{f}_R^\dagger P^{-1} H_E P \vec{f}_L}\right) P_{n,\alpha}^{-1}, \tag{51}$$

where $\vec{f}_R^\dagger = \vec{c}^\dagger P$ and $\vec{f}_L = P^{-1}\vec{c}$, the non-Hermitian Gaussian state defined by the entanglement Hamiltonian

$$H_E = PQ^{-1}J(\log\left[s^{-1}-1\right])QP^{-1}, \tag{52}$$

has $C$ for its correlation matrix. Due to the matrix $P$, the basis in which $H_E$ takes a Jordan form is not the same as in $C$.

## B  Inverse participation ratio

In this Section, we introduce the definitions of the inverse participation ratio (IPR) we use in the main text to visualize the spatial support of the eigenmodes of the entanglement Hamiltonian. In a Hermitian context, it is defined as follows

$$IPR(|R_n\rangle) = \frac{\left(\sum_{j,\sigma=A/B} |\langle j,\sigma|R_n\rangle|^2\right)^2}{\sum_{j,\sigma=A/B} |\langle j,\sigma|R_n\rangle|^4}, \tag{53}$$

where $\{|j,\sigma\rangle\}$ is the (canonic) real space basis of the single-particle Hilbert space, where $j$ denotes the unit-cell and $\sigma = A/B$ the sublattice. The inverse participation ratio estimates the support of the mode $|R_n\rangle$ in the basis $\{|j\rangle\}$: It is equal to 1 for a perfectly localized state on a single site, and $2l$ for a state fully delocalized on $l$ unit-cells and both sublattices. We use this definition for the eigenstates of the right entanglement Hamiltonian.

When using the biorthogonal formulation of quantum mechanics, we evaluate observables by computing

$$\langle\mathcal{O}\rangle_{RL} = \langle\phi^L|\mathcal{O}|\phi^R\rangle. \tag{54}$$

We are therefore interested more in the (bi)localization of the product $|\phi^L\rangle$ and $|\phi^R\rangle$, i.e. in the localization of $\langle n_{j,\sigma}\rangle_{RL}$. It is therefore more coherent to study the ratio

$$IPR^{RL}(|R_n\rangle) = \frac{\left(\sum_{j,\sigma=A/B} |\langle L_n|j,\sigma\rangle\langle j,\sigma|R_n\rangle|\right)^2}{\sum_{j,\sigma=A/B} |\langle L_n|j,\sigma\rangle\langle j,\sigma|R_n\rangle|^2}. \tag{55}$$

It coincides then with localization of the expectation values $\langle n_j\rangle = \langle n_{j,A}\rangle + \langle n_{j,B}\rangle$ of the corresponding many-body wave-function, as defined in Eq. (36).

Finally, when studying the singular value decomposition of the entanglement Hamiltonian $H_E = U\Lambda V^\dagger$, we choose for similar reasons

$$IPR^{SVD}(|U_n\rangle) = \frac{\left(\sum_{j,\sigma=A/B} |\langle V_n|j,\sigma\rangle\langle j,\sigma|U_n\rangle|\right)^2}{\sum_{j,\sigma=A/B} |\langle V_n|j,\sigma\rangle\langle j,\sigma|U_n\rangle|^2}, \tag{56}$$

where $|U_n\rangle$ ($|V_n\rangle$) is the $n^{\text{th}}$ column of $U$ ($V$) respectively.

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
