# Peer review of "Entanglement spectrum and symmetries in non-Hermitian fermionic non-interacting models"

_SciPost Physics, doi:SciPost Phys. 7, 069 (2019)_

## Round 1 · Referee Report · Gunnar Moller · 2019-10-9

Strengths

1- good review of background information regarding the Hamiltonian and density matrix for both Hermitian and non-Hermitian systems
2- convincing extension of the definition of entanglement spectrum for non-Hermitian systems, for the two possible definitions of reduced density matrices
3- generalisation of Wick's theorem and the corresponding extension of how to calculate the entanglement of free fermions from the correlation matrix of non-Hermitian Hamiltonians
4- comprehensive classification of possible entanglement spectra as a function of the non-Hermitian symmetries present in the original Hamiltonian
5- range of specific examples, including both analytical and numerical studies

Weaknesses

1- The paper does not offer many illustrations of the complex relations between the possible symmetries. Reading could be facilitated if the authors add an illustration of the symmetry and generic structure of the entanglement spectrum, at least for some of the examples discussed in Section IV.

Report

This is an excellent paper, which introduces and studies two different notions of entanglement spectrum for non-Hermitian systems. The two possible definitions of ES arise naturally from the alternative use of left and/or right eigenvectors in constructing the system's density matrix.
The analytical results (see strenghts) are convincingly derived, and backed up by the study of a range of example systems. Thanks to their focus on non-interacting systems, the authors can study large system sizes, which display the features of the non-Hermitian ES. The authors believe that these will essentially carry over to interacting systems, which would imply that the results will enjoy significant importance in a range of different physical systems.

The scientific aspects are clearly sound, so this paper has my strong recommendation.
I have one minor comment about a loosely worded sentence in the conclusions, and a list of minor changes, given below.

Requested changes

1- In their conclusions, the authors should clarify the following sentence: "It appears then, that the bulk-boundary correspondence holds for the ES in line-gapped Hamiltonians, when considering the bi-orthogonal density matrix." How strong can this point be made, and if needed, what qualifications of this statement apply?

2- No comment is given to whether the locations of phase transitions seen for the singular values and eigenvalues agree in Figures 6 and 8. Given the scale, it is difficult to discern if they coincide, so could the authors add some vertical lines as visual guides to clarify the location of the other transition, respectively, in the different panels. Some discussion of this point would also be helpful.

3- In equation (3) and thereafter, the typesetting of $\vec c^\dagger$ requires some attention, as the vector and dagger signs overlap.

4- On page, three, below (12), the authors refer to equation (8), where I believe the reference should be to (10). This referencing error is repeated below (14).

5- On page 4, discussing the case of non-diagonalisable correlation matrices, the authors assert that the Jordan blocks of $H_E$ and $C$ are identical, while their form bases differ. Please expand on this assertion, if needed within an appendix.

6- The authors should point out explicitly that the resulting entanglement spectrum (25) follows the same relation to the correlation spectrum as in the Hermitian case.

7- On page 8, the sentence "[...] symmetries do not carry on to the right density matrix [...]" is unclear.

  • validity: top
  • significance: high
  • originality: high
  • clarity: high
  • formatting: excellent
  • grammar: excellent

Author:  Loïc Herviou  on 2019-10-18  [id 629]

(in reply to Report 1 by Gunnar Moller on 2019-10-09)
Category:
answer to question
correction

Dear Editor and Referee,

We would like first to thank the referee for his very positive report. We address below the different changes requested by the referee.

1- In their conclusions, the authors should clarify the following sentence: "It appears then, that the bulk-boundary correspondence holds for the ES in line-gapped Hamiltonians, when considering the biorthogonal density matrix." How strong can this point be made, and if needed, what qualifications of this statement apply?

Reply: We do not have a strict analytical proof of this statement, so we cannot make it stronger than it currently is: it is a conjecture based on strong numerical evidence. More precisely, we have shown that the entanglement Hamiltonian belongs to the same topological class as the original Hamiltonian under a set of symmetry-dependent assumptions on which eigenstates are occupied. These assumptions are generically compatible with having line gaps (for example the absence of strictly imaginary (alternatively real) eigenmodes in some symmetry classes).

We have not directly proven though that the entanglement Hamiltonian has to have the same topological invariants as the original Hamiltonian in that case. For simple symmetry classes (winding numbers/Chern numbers), the usual proof in the Hermitian case should apply, but we have not deeply investigated the question. This is tightly related to a property of line-gapped Hamiltonians: according to Kawabata et al., they can be smoothly deformed into Hermitian models, for which we know that the correspondence holds.

We developed the discussion in the conclusion and replaced the sentence: ""It appears then, that the bulk-boundary correspondence holds for the ES in line-gapped Hamiltonians, when considering the bi-orthogonal density matrix." by "It appears then, that the bulk-boundary correspondence holds for the biorthogonal entanglement spectrum in line-gapped Hamiltonians. We have indeed analytically shown that the entanglement Hamiltonian belongs to the same symmetry class as the original Hamiltonian, under symmetry dependent assumptions (e.g., the absence of purely imaginary eigenmodes) which are generically valid in line-gapped phases, and numerically shown that their corresponding topological invariants then match. This result can be seen as a consequence of the fact that line-gapped non-Hermitian Hamiltonians are smoothly deformable into (anti-)Hermitian ones[43], for which we know the bulk-boundary correspondence generally holds."

2- No comment is given to whether the locations of phase transitions seen for the singular values and eigenvalues agree in Figures 6 and 8. Given the scale, it is difficult to discern if they coincide, so could the authors add some vertical lines as visual guides to clarify the location of the other transition, respectively, in the different panels. Some discussion of this point would also be helpful.

Reply: In both cases, the observed phase transitions occur simultaneously for both eigenvalues and singular values. Following the referee's suggestion, we added vertical lines marking the phase transition in Figs. 6 and 8. We adjusted the captions accordingly, and added the mention of this result in the main text. We also changed the style of Fig. 3 to exactly match Figs. 6 and 8, and also introduced the corresponding vertical lines.

3- In equation (3) and thereafter, the typesetting of \vec{c}^\dagger requires some attention, as the vector and dagger signs overlap.

Reply: We adjusted the spacing so that the vector arrow and the dagger symbol are clearly separated.

4- On page, three, below (12), the authors refer to equation (8), where I believe the reference should be to (10). This referencing error is repeated below (14).

Reply: The reference was indeed incorrect. Thank you for pointing it out.

5- On page 4, discussing the case of non-diagonalisable correlation matrices, the authors assert that the Jordan blocks of HE and C are identical, while their form bases differ. Please expand on this assertion, if needed within an appendix.

Reply: We clarified the discussion of the Jordan form canonical basis in the main text:

"If the correlation matrix is diagonalizable, the corresponding entanglement Hamiltonian is also diagonalizable, and its eigenmodes are the eigenvectors of the correlation matrix. If the correlation matrix is not diagonalizable, the entanglement Hamiltonian $H_E$ is also not diagonalizable. It follows naturally from App. A that the correlation matrix and the entanglement Hamiltonian have the same Jordan block structure (same number of Jordan blocks of the same size). Matching Jordan blocks in the correlation matrix and the entanglement Hamiltonian act on the same eigenspace. The canonical basis of this eigenspace that leads to the Jordan form will generically be different in the two matrices."

All the elements of proof of these statements were already present in App. A. We also added the two following sentences to clarify our proof. "Each eigenspace (corresponding to independent Jordan blocks) can be treated separately." "Due to the matrix $P$, the basis in which $H_E$ takes a Jordan form is not the same as in $C$."

6- The authors should point out explicitly that the resulting entanglement spectrum (25) follows the same relation to the correlation spectrum as in the Hermitian case.

Reply: We added the following sentence at the end of Section III "Note that the formula (25) is the same as in the Hermitian case."

7- On page 8, the sentence "[...] symmetries do not carry on to the right density matrix [...]" is unclear.

Reply: We clarified our assertion and replaced "In particular, the $T_-$ and $P_+$ symmetries do not carry on the right density matrix even though they are relevant to the line gap classification." by "In particular, while the $T_-$ and $P_+$ symmetries are relevant to the line gap classification, they only map the right density matrix $\rho^R$ to the left density matrix $\rho^L$, which does not put any strong constraints on $rho^R$ itself, and therefore does not constrain its topological properties."

Additional minor change: - we added Ref. 87 at the beginning of Sec VII A - We added the following sentence, referencing a related work published on arXiv shortly after ours. "Following this work, a similar approach was developed in Ref. 92 to study the entanglement entropy and therefore the effective low-energy conformal field theory describing critical points in non-Hermitian models."

---

## Round 3 · Referee Report · Gunnar Moller (Referee 1) · 2019-11-7

Strengths

several, see previous report

Weaknesses

no significant weaknesses, see previous report

Report

The authors have addressed all of my criticisms very well, so I can warmly recommend the paper in its current form.

I also thank the authors for adding a reference to a related work published on this topic.

Requested changes

None.

---

## Round 3 · Author Response

Most changes follow from the referee reports

---

## Round 3 · List of Changes

Changes to the main text - Page 3: corrected wrong reference to Eq. (10) - In Sec III: clarified discussion on Jordan blocks "If the correlation matrix is diagonalizable, [...] will generically be different in the two matrices." - In Sec III: added the sentence "Note that the formula (25) is the same as in the Hermitian case." - In Sec IVB: clarified our assertion and replaced "In particular, the $T_-$ and $P_+$ symmetries[...] line gap classification." by "In particular [...] its topological properties." - In Sec VI, we clarified that singular values and eigenvalues undergoes transition at the same moment, for Fig.3, 6 and 8. - Added Ref. 87 at the beginning of Sec VII A - In Conclusions: developed the discussionon the bulk-boundary correspondence and replaced ""It appears [...] density matrix." by "It appears [...] bulk-boundary correspondence generally holds."

  • Added the following sentence, referencing a related work published on arXiv shortly after ours. "Following this work, a similar approach was developed in Ref. 92 to study the entanglement entropy and therefore the effective low-energy conformal field theory describing critical points in non-Hermitian models."

Update to the acknowledgements - Added "N.R. was supported by [...]Memorial Foundation."

Change in Figures - Updated Fig. 3, 6 and 8 and their caption to underline when the entanglement spectrum undergoes a phase transition

Change in Appendix -Added the following two sentences to App. A "Each eigenspace (corresponding to independent Jordan blocks) can be treated separately." "Due to the matrix $P$, the basis in which $H_E$ takes a Jordan form is not the same as in $C$."

Formatting changes - changed the spacing from \vec{c}^\dagger to \vec{c}^{\: \dagger} - Appendix to App. when relevant

---

## Editorial Decision

published